

# Long-range atmospheric transport of volatile monocarboxylic acids with Asian dust over high mountain snow site, central Japan

Tomoki Mochizuki[1,2], Kimitaka Kawamura[1,3*], Kazuma Aoki[4], and Nobuo Sugimoto[5]

[1]Institute of Low Temperature Science, Hokkaido University, N19 W8, Kita-ku Sapporo, 060-0819, Japan
5 [2]Now at School of Food and Nutritional Sciences, University of Shizuoka, 52-1 Yada, Shizuoka 422-8526, Japan
[3]Now at Chubu Institute for Advanced Studies, Chubu University, 1200 Matsumoto-cho, Kasugai 487-8501, Japan
[4]Department of Earth Science, Faculty of Science, University of Toyama, 3190, Gofuku, Toyama 930-8555, Japan
[5]National Institute of Environmental Studies, 16-2 Onogawa, Tsukuba, Ibaraki 305-8506, Japan

*Correspondence to*: K. Kawamura (kkawamura@ isc.chubu.ac.jp)

10 **Abstract.** To understand the long-range transport of monocarboxylic acids from the Asian Continent to Japanese Islands, we collected snowpack samples from a pit sequence (depth, ca. 6 m) at the Murodo-Daira snowfield near the summit of Mt. Tateyama, central Japan in 2009 and 2011. Snow samples (n=16) were analyzed for normal ($C_1$-$C_{10}$), branched chain ($iC_4$-$iC_6$), aromatic (benzoic and toluic acid isomers), and hydroxyl (glycolic and lactic) monocarboxylic acids, together with inorganic ions and dissolved organic carbon (DOC). Acetic acid ($C_2$) was found as a dominant species (average, 125 ng $g^{-1}$), 15 followed by formic acid ($C_1$) (85.7 ng $g^{-1}$) and isopentanoic acid ($iC_5$) (20.0 ng $g^{-1}$). We found a strong correlation ($r^2$=0.73) between formic plus acetic acids and non-sea-salt $Ca^{2+}$ that is a proxy of Asian dust. Contributions of total monocarboxylic acids to DOC in 2009 (21.2±11.6%) were higher than that in 2011 (3.75±2.62%), being consistent with higher intensity of Asian dust in 2009 than 2011. Formic plus acetic acids also showed a positive correlation ($r^2$=0.81) with benzoic acid that is a tracer of automobile exhaust, indicating that monocarboxylic acids and their precursors are largely emitted from 20 anthropogenic sources in China and/or secondarily produced in the atmosphere by photochemical processing. In addition, the ratio of formic plus acetic acids to nss-$Ca^{2+}$ (0.27) was significantly higher than those (0.00036-0.0018) obtained for reference dust materials of Chinese loess deposits from Tengger and Gobi deserts. This result suggests that volatile and semi-volatile organic acids are adsorbed on the alkaline dust particles during long-range atmospheric transport. Entrainment of organic acids by dusts is supported by a good correlation ($r^2$=0.76) between formic plus acetic acids and pH of melt snow 25 samples. Our study demonstrates that Asian alkaline dusts can uptake volatile monocarboxylic acids during long-range transport and the dusts coated with organic acids may act as effective ice nuclei to cause a heavy snowfall.

## 1 Introduction

Low molecular weight (LMW) monocarboxylic acids such as formic (HCOOH) and acetic ($CH_3COOH$) acids are present in the atmosphere as major gaseous and particulate organic components (e.g., Kawamura and Kaplan, 1984; Kawamura et al., 2000; Paulot et al., 2011). They have been reported in urban (Kawamura et al., 2000) forest (Andreae et al., 1988), high



mountain (Preunkert et al., 2007), marine (Miyazaki et al., 2014), and Arctic samples (Talbot et al., 1992; Legrand et al., 2004). Salts of organic acids in aerosols are water-soluble and thus influence the radiation budget of the earth's atmosphere by acting as cloud condensation nuclei (CCN) (Kanakidou et al., 2005). In addition, LMW monocarboxylic acids have been detected in wet deposition such as rain, cloud and fog water, and snow samples (Maupetit and Delmas, 1994; Keene et al.,

1995; Kawamura et al., 1996; Kawamura et al., 2012). Thus, organic acids are scavenged by wet deposition from the upper troposphere as CCN and ice nuclei (IN). It is also important to note that organic acids largely contribute to total acidity of rainwaters (Kawamura et al., 1996; Keen et al., 1983).

Formic and acetic acids have variety sources such as primary emission from motor exhausts (Kawamura et al., 2000) and vegetation (Kesselmeier et al., 1998), and secondary formation via the oxidation of anthropogenic and biogenic precursors

such as toluene (Mochizuki et al., laboratory data) and isoprene (Paulot et al., 2011). Kawamura et al. (2000) reported that photochemical oxidations of various organic precursors are more important as a source of monocarboxylic acids in the troposphere. On the other hand, microorganisms are known to produce branched chain ($iC_4$ and $iC_5$) monocarboxylic acids (Allison, 1978).

The Japanese Islands are located in the western North Pacific Rim, which is influenced by the Asian outflow of dusts and air

pollutants. Asian dust (Kosa) events in the desert areas of North China promote the delivery of air pollutants with dust particles to the western North Pacific by westerly winds (e.g., Iwasaka et al., 1983; Huebert et al., 2003). LMW monocarboxylic acids have been detected in alpine snow samples collected near the summit of Mt. Tateyama (Kawamura et al., 2012). They reported higher concentrations of monocarboxylic acids in snow pit samples with dust layers, suggesting that monocarboxylic acids may be associated with Asian dust during long-range atmospheric transport.

During the winter monsoon season, the Japanese high mountains facing to the Sea of Japan are known to have a heavy snowfall, which is associated with a significant evaporation of water vapours from the warm Tsushima Current in the Sea of Japan under a strong westerly wind condition. Because these snow precipitations are the sources for agricultural and drinking waters in those areas, snow precipitations in the high mountains are important for the water cycles in Japan. Ice nuclei (IN) supplied from the Asian Continent through long-range atmospheric transport and the water vapours supplied from the Sea of

Japan during the winter Asian Monsoon are two important components to promote the heavy snow over the western part of Japan. Asian dust particles may act as effective IN to result in a heavy snow over the Japanese Alps. During the formation of snow crystals, chemical compositions of dust surface may be critical to control the hygroscopic properties of dust particles (Creamean et al., 2013). Alpine mountain snow sequences would provide useful information on the chemical states of Asian dust deposited over the snowfield, in which atmospheric organic acids are well preserved in snow layers (Osada et al., 2004).

In the present study, we collected snowpack samples from a pit sequence in the Murodo-Daira snowfield (ca. 6 m in depth) near the summit of Mt. Tateyama, central Japan in April of 2009 and 2011. To better understand the sources of monocarboxylic acid and their long-range transport by Asian dust over the Japanese Islands, 16 snowpack samples were analyzed for monocarboxylic acids, inorganic ions, and dissolved organic carbon (DOC) as well as reference dust materials of Chinese loess deposit samples collected from the Tengger and Gobi deserts. We discuss the contributions of LMW



monocarboxylic acids to DOC as well as the association of monocarboxylic acids with alkaline dust particles during long-range atmospheric transport. Relations between monocarboxylic acids and pH values of the snow melt water will also be discussed in terms of atmospheric titration of alkaline dust particles by acidic species including organic acids during atmospheric transport.

## 2 Material and methods

### 2.1 Sample collection

The details on the snow collection and sample storage methods were described in Kawamura et al. (2012) and Mochizuki et al. (2016). Snowpack samples were collected at the Murodo-Daira site (36.58º N, 137.36º E; elevation, 2450 m) near Mt. Tateyama (elevation, 3015 m), central Japan (Fig. 1). Snow pit hole (depth, ca. 6 m) was dug down to the ground. Table 1 provides descriptions of snow samples collected from the snow pit sequence, in which several dirty layers were recognized by visual observation due to the occurrence of Asian dusts. Five snowpack samples including three dust layers were collected from the pit sequence in April 18, 2009. Eleven snowpack samples including four dust layers were collected from the pit sequence in April 17, 2011. In order to evaluate the homogeneity of snow samples within the same snow horizon with dust layer, another snowpack sample (#4') was collected at ca. 1 m away from the location of sample #4 in 2009. Because the thickness of dust layers in snow pit sequence is ca. 10 cm or more, dusts are deposited together with snowflakes during snow precipitation rather than dry deposition.

The snow samples were placed in a pre-cleaned glass jar (8 L) using a clean stainless steel scoop. To avoid microbial degradation of organic compounds, mercuric chloride ($HgCl_2$) was added to the glass jar prior to collect snow sample. The sample jars were sealed with a Teflon-lined screw cap and transported to the laboratory in Sapporo within four days by a commercial refrigerated transport service, which kept the samples in darkness at ca. 5 °C and constant humidity. The samples were stored in a dark refrigerator room at 4 °C prior to analysis.

We also analysed the reference dust materials (Kosa) including Chinese loess deposits from the Tengger desert (CJ-1, < 250 µm and CJ-2, < 100 µm), and Gobi desert (Gobi, < 10 µm). The reference materials were purchased from the National Institute for Environmental Studies, for the measurements of LMW monocarboxylic acids, inorganic ions, and DOC. Reference dust samples (0.1 g) were extracted with ultra pure water by the methods as described below. The data of inorganic ions and trace elements in the reference samples are reported elsewhere (Nishikawa et al., 2000, 2013).

### 2.2 Organic chemical analysis

Monocarboxylic acids were determined as p-bromophenacyl esters using capillary gas chromatography (GC) and GC-mass spectrometry (GC-MS) methods (Kawamura and Kaplan, 1984). 150 ml of melted snow samples were transferred to a pear-shape glass flask (300 ml) and the pH was adjusted to 8.5–9.0 with 0.05 M KOH solution. The sample was concentrated down to 10 ml using a rotary evaporator under vacuum at 50 °C. The concentrates were filtered through quartz wool packed



in a Pasteur pipette. The filtrates were concentrated down to 0.5 ml. To convert all organic acids to RCOO⁻K⁺ form, the concentrates were passed through a glass column (Pasteur pipette) packed with cation exchange resin (DOWEX 50W-X4, 100-200 mesh, K⁺ form). Organic acids were eluted with pure water and transferred in a 25 ml pear-shape flask. The pH of the sample was checked to be 8.5–9.0 and then dried using a rotary evaporator under vacuum, followed by blown-down with

pure nitrogen gas.

Acetonitrile (4 ml) was added to the dried sample, and RCOO⁻K⁺ salts were reacted with α,p-dibromoacetophenone (0.1 M, 50 μl) as a derivatization reagent and dicyclohexyl-18-crown-6 (0.01 M, 50 μl) as a catalyst to derive p-bromophenacyl esters at 80 °C for 2 hours. The reaction mixture was dried using a rotary evaporator under vacuum at 30 °C. The derived esters were dissolved in 0.5 ml of n-hexane/dichloromethane (2:1) mixture and then purified on a silica gel column (Pasteur

pipette). Excess reagent was eluted with n-hexane/dichloromethane (2:1) mixture (7 ml) and then p-bromophenacyl esters were eluted with dichloromethane/methanol (95:5) mixture (2 ml) into a glass vial (2 ml). The esters were dried by blown down using pure nitrogen gas and then dissolved in n-hexane (100 μl). In addition, the esters of hydroxyacids (lactic and glycolic acids) were reacted with N,O-bis-(trimethylsilyl) trifluoroacetamide (BSTFA) with 1% trimethylsilyl chloride and 10 μl of pyridine to derive trimethylsilyl (TMS) ethers for hydroxyl (OH) group at 70 °C for 3 hours.

p-Bromophenacyl esters and their TMS ethers were determined using a capillary gas chromatograph (HP GC6890, Hewlett-Packard, USA) equipped with a flame ionization detector. The esters were separated using a fused silica capillary column (HP-5, 30 m × 0.2 mm i.d., film thickness 0.5 μm). The derivatives were also analysed by GC-MS (Agilent GC7890A and 5975C MSD, Agilent, USA). The compounds were identified by comparing GC retention time and mass spectra of authentic standards. Details of analytical procedure were described previously (Kawamura et al., 2012). We tested the recoveries of

authentic monocarboxylic acid standards ($C_1$-$C_{10}$, $iC_4$-$iC_6$, benzoic, toluic, lactic, and glycolic acids) that were spiked into ultra pure water. The results showed that the recoveries of organic acids were better than 80%. Analytical errors using authentic standards were within 12%. Detection limits of organic acids were estimated to be 0.001–0.004 ng g⁻¹.

To measure inorganic ions, samples were passed through a membrane disc filter (0.22 μm, Millipore Millex-GV, Merck, USA) and the filtrates were injected to an ion chromatograph (Model 761 compact IC, Metrohm, Switzerland) equipped with

an AS-09 autosampler (Kawamura et al., 2012). Anion analysis was conducted using a Shodex SI-90 4E column and a 1.8 mM $Na_2CO_3$ + 1.7 mM $NaHCO_3$ solution as eluent. Cation analysis was conducted using a C2-150 column and a 4.0 mM tartaric acid + 1.0 mM dipicolinic acid solution as eluent. The total analytical precision is 4% (Miyazaki et al., 2010).

After removing the particles in the samples on a disk filter (0.22 μm, Millipore Millex-GV, Merck, USA), dissolved organic carbon (DOC) was determined using a total organic carbon (TOC) analyzer (Model TOC-Vcsh, Shimadzu) (Miyazaki et al.,

30    2011).

## 2.3 Non-sea salt ions

Concentrations of non-sea salt ionic species X ($M_{nss-x}$) were estimated by the following equation:

$M_{nss-x} = M_x - (X/Na)_{sw}M_{Na}$





where $M_x$ and $M_{Na}$ are the concentrations of X and of Na, respectively. $(X/Na)_{sw}$ means mass ratio of species X to Na in seawater (Duce et al., 1983). The ratios are 0.25 ($SO_4^{2-}$), 0.037 ($K^+$), 0.038 ($Ca^{2+}$), and 0.12 ($Mg^{2+}$) (Berg and Winchester, 1978). The ratio of $F^-$ is 0.000146 (Yang et al., 2009).

**2.4 Lidar observation and back trajectory analysis**

We detected Asian dust events by the lidar observation (http://www-lidar.nies.go.jp/Toyama/) over Imizu (36.70º N, 137.10º E), ca. 40 km northwest of Mt. Tateyama, Toyama Prefecture, Japan during December to March in each year. The observation wavelength of laser is 532 nm. Details of the extinction coefficient measurements of dust particles were described in Shimizu et al. (2004). One example of lidar image is presented in Figure 2. Dense dust layers were recorded at the upper layers (3-4 km) over Imizu during December 10, 2008, whose dust event should be recorded in the snow pit sequences collected in 2009 (possibly corresponds to #4, see Table 1). This dust event was also recognized by the lidar observations at Niigata, Sendai, and Tsukuba in Japan. We estimated that Asian dust events observed in December 10, January 1 and February 2 during 2008-2009 and December 6, 25-26 and 31 and February 22-24 during 2010-2011 correspond to sample ID #4, #3, #1, #14, #12, #10 and #7, respectively (Table 1).

To investigate the source of air masses during snow season (November to April), seven-day backward air mass trajectories were calculated at a level of 3000 m a.s.l. using online program, Meteorological Data Explorer (METEX), which was developed by the National Institute for Environmental Studies (NIES), Japan. Meteorological data were obtained from the National Centers for Environmental Prediction (NCEP) Reanalysis data. Figure 3 shows the back air mass trajectories corresponding to selected dust layers (Table 1). The heights of air masses over the Asian Continent and the Sea of Japan ranged from 2500 m to 6000 m.

**3 Results**

**3.1 Tateyama snow pit samples**

Homologous series of low molecular weight normal aliphatic ($C_1$-$C_{10}$), branched chain ($iC_4$-$iC_6$), hydroxy (lactic and glycolic), and aromatic (benzoic acid and o-, m- and p-toluic acid isomers) monocarboxylic acids were detected in the snow pit samples (Table 2). We found that differences in the concentrations of each monocarboxylic acids between sample #4 and sample #4' are within the analytical uncertainties. Thus, we consider that each horizontal layers in the snow pit site are homogenous and each snow samples are representative of the snowfall events over the Murodo site.

Acetic acid ($C_2$) was found as dominant species (2009: 51.2–708 ng g$^{-1}$, 2011: 9.01–61.5 ng g$^{-1}$) followed by formic acid ($C_1$) (2009: 41.8–476 ng g$^{-1}$, 2011: 2.21–62.0 ng g$^{-1}$). Concentrations of $C_3$-$C_{10}$ acids were 1–2 orders of magnitude lower than $C_2$. In contrast, $iC_5$ acid (2009: 30.6–114 ng g$^{-1}$, 2011: 0.55–3.66 ng g$^{-1}$) was detected as the most abundant branched chain acid. Lactic and glycolic acids were also detected as hydroxyacids in the snow pit samples. However, concentrations of lactic and glycolic acids are 1 and 2 orders of magnitude lower than those of major monocarboxylic acids ($C_1$ and $C_2$), respectively.



Concentration of benzoic acid ranged from 0.08 to 8.74 ng $g^{-1}$. Total concentrations of toluic acid isomers were found to be significantly lower (average, 0.07 ng $g^{-1}$) than that of benzoic acid (2.11 ng $g^{-1}$). Higher concentrations of monocarboxylic acids were observed in the snow samples with the dust layers than those without dust layers in both 2009 and 2011. Concentrations of dissolved organic carbon (DOC) ranged from 469 ng $g^{-1}$ to 2380 ng $g^{-1}$ in 2009 and 381 ng $g^{-1}$ to 2110 ng

$g^{-1}$ in 2011 (Table 2). The highest concentration of DOC (2380 ng $g^{-1}$) was found in sample #3, in which dust layer was observed.

We detected cations ($Ca^{2+}$, $Na^+$, $Mg^{2+}$, $K^+$, and $NH_4^+$) and anions ($F^-$, $NO_3^-$, $SO_4^{2-}$, and $MSA^-$) in snow pit samples collected in both 2009 and 2011 from the Murodo-Daira site near Mt. Tateyama (Table 3). Concentrations of nss-$Ca^{2+}$, nss-$Mg^{2+}$, nss-$K^+$, nss-$F^-$ and nss-$SO_4^{2-}$ were calculated as shown in Table 3. $NO_3^-$ and nss-$SO_4^{2-}$ are two major anions. The highest

concentrations of $NO_3^-$ (2020 ng $g^{-1}$) and nss-$SO_4^{2-}$ (2440 ng $g^{-1}$) were obtained in sample #10, in which a dirty dust layer was observed. On the other hand, $Na^+$ and nss-$Ca^{2+}$ are two major cations. Higher concentrations of $Na^+$ and nss-$Ca^{2+}$ were found in samples #1 ($Na^+$: 3240 ng $g^{-1}$; nss-$Ca^{2+}$: 3000 ng $g^{-1}$) and #3 ($Na^+$: 5210 ng $g^{-1}$; nss-$Ca^{2+}$: 3190 ng $g^{-1}$), both of which showed the presence of dust layer.

The pH of melt snow samples ranged from 4.4 to 6.9 (Table 3). The higher pH were found in samples #1, #3, and #4 (pH =

6.7–6.9), in which dust layers were observed.

Average concentrations of $C_1$ (202 ± 170 ng $g^{-1}$), $C_2$ (292 ± 249 ng $g^{-1}$), and $iC_5$ (53.5 ± 30.8 ng $g^{-1}$) in 2009 are one order of magnitude higher than those in 2011 ($C_1$: 22.4 ± 20.1 ng $g^{-1}$; $C_2$: 34.2 ± 15.8 ng $g^{-1}$; $iC_5$: 1.69 ± 0.88 ng $g^{-1}$). Similar tends were found for the average concentrations of minor monocarboxylic acids ($C_3$-$C_{10}$, $iC_4$ and $iC_6$) in the snow pit samples in 2009 and 2011. Average concentration of DOC in 2009 (1090 ± 712 ng $g^{-1}$) is slightly higher than that in 2011 (836 ± 534 ng

$g^{-1}$). Interestingly, the contribution of total monocarboxylic acids to DOC (Total MCA-C/DOC) in 2009 (21.2±11.6%) is 6 times higher than that in 2011 (3.75±2.62%).

Average concentrations of $NO_3^-$ (657 ± 633 ng $g^{-1}$) and nss-$SO_4^{2-}$ (748 ± 682 ng $g^{-1}$) in 2011 are 2-3 times higher than those in 2009 ($NO_3^-$: 302 ± 166 ng $g^{-1}$; nss-$SO_4^{2-}$: 207 ± 139 ng $g^{-1}$). In contrast, average concentrations of nss-$Ca^{2+}$ in 2009 (1740 ± 1190 ng $g^{-1}$) are 5 times higher than those in 2011 (345 ± 285 ng $g^{-1}$).

**3.2 Reference dust materials**

We detected LMW monocarboxylic acids, inorganic ions, and DOC in the water extracts from three reference dust materials (CJ-1, CJ-2 and Gobi). Concentrations of total LMW monocarboxylic acids in the reference dusts were 4,370 ng $g^{-1}$ (CJ-1), 29,390 ng $g^{-1}$ (CJ-2), and 21,010 ng $g^{-1}$ (Gobi). The dominant LMW monocarboxylic acids were formic and acetic acids. Concentrations of DOC were 73,000 ng $g^{-1}$ (CJ-1), 403,000 ng $g^{-1}$ (CJ-2), and 267,000 ng $g^{-1}$ (Gobi). Total MCA-C/DOC

ratios in reference dust materials were 2.0% (CJ-1), 2.9% (CJ-2), and 3.3% (Gobi). Concentrations of nss-$Ca^{2+}$ in the reference dust materials were 10,700 µg $g^{-1}$ (CJ-1), 18,700 µg $g^{-1}$ (CJ-2), and 8,820 µg $g^{-1}$ (Gobi).



## 4 Discussion

### 4.1 Influence of Asian dust

High concentrations of nss-$Ca^{2+}$ were obtained in the dust layers of both 2009 and 2011. $Ca^{2+}$ is known as a major metal ion to be transported from arid regions in North Asia with Asian dust (Mori et al., 2002; Tsai and Chen, 2006). In this study, contributions of nss-$Ca^{2+}$ to $Ca^{2+}$ in 2009 and 2011 are 95% and 91%, respectively. In addition, the ratios of Mg/Ca at the Murodo-Daira site in 2009 and 2011 are 0.08 and 0.12, respectively. These values are comparable to those in reference dust materials such as CJ-1 (0.17), CJ-2 (0.06) and Gobi (0.09). Therefore, nss-$Ca^{2+}$ can be used as an indicator of mineral dust. High abundances of nss-$Ca^{2+}$ in snowpack samples indicate that a strong outflow of dust particles from the Asian Continent has involved with a heavy snow precipitation.

To investigate the effect of Asian dust on LMW monocarboxylic acids, we plotted major LMW monocarboxylic acids (i.e., formic plus acetic acids) against nss-$Ca^{2+}$ using all the data points (Fig. 4). We found that formic plus acetic acids strongly correlate with nss-$Ca^{2+}$ ($r^2 = 0.73$). The air mass trajectories have passed over the Asian Continent including North China and Mongolia (Fig. 3). The atmospheric transport of Asian dust over the Japanese Islands is a dominant factor to control the concentrations of nss-$Ca^{2+}$ and formic and acetic acids in snow precipitations during winter to spring, when the Asian dust activity maximizes in North China. The pathways of long-range transport and sources of formic and acetic acids will be discussed in the following sections 4.2 and 4.3.

Average concentrations of formic and acetic acids and nss-$Ca^{2+}$ in 2009 are higher than those in 2011. Total rainfall during December to February in Urumqi near the Taklamakan desert, China in 2009 (31 mm) was half of that in 2011 (60 mm) (Japan Meteorological Agency website: http://www.data.jma.go.jp/gmd/cpd/monitor/mainstn/obslist.php), suggesting that soil moisture in 2009 should have been lower than that in 2011 and thus the soil surfaces in 2009 should have been more dried. The higher concentrations of organic acids and nss-$Ca^{2+}$ in the 2009 snow pit samples should be caused by a strong influence of the Asian dust events in this year when the soil surfaces were more dried in the arid regions, although the detailed records of the Asian dust events in North China are not available.

### 4.2 Long-range transport of formic and acetic acids and aerosol acidity/alkalinity

Although the alkalinity snow pit samples can be affected by titration of alkaline dust particles, melt snow samples from the Murodo-Daira were slightly acidic. Figure 5 presents the relationship between formic plus acetic acids and pH of melt snow. We found that formic plus acetic acids positively correlated with pH ($r^2 = 0.76$). Their concentrations exponentially increased at pH range of > 6.0. Interestingly, concentrations of nss-$Ca^{2+}$ positively correlate with pH ($r^2 = 0.79$) (Fig. 5). Because LMW monocarboxylic acids have high vapour pressure (Saxena and Hildeman, 1996), they should be largely present as gases in the atmosphere (e.g., Kawamura et al., 1985; Liu et al., 2012). During long-range atmospheric transport, alkaline dust particles may be subjected to atmospheric titration by gaseous monocarboxylic acids. In contrast, $SO_4^{2-}$ and $NO_3^-$ did not show any correlation with pH ($r < 0.10$). These results may suggest that, during the atmospheric transport, dust





particles do not efficiently uptake inorganic acids that are mostly present as particles. The adsorption process of acidic components by aerosols may be different between gaseous organic acids and inorganic gases such as $SO_2$ and $NO_x$, which are precursors of $SO_4^{2-}$ and $NO_3^-$, respectively.

We calculated the ratios of formic plus acetic acids/nss-$Ca^{2+}$ for the Murodo-Daira snowpit samples and compared the ratios

of formic plus acetic acids/nss-$Ca^{2+}$ in the reference materials such as CJ-1, CJ-2 and Gobi. We found that formic plus acetic acids/nss-$Ca^{2+}$ ratios for the Murodo-Daira snowpit samples (av. 0.27) are significantly higher than those from CJ-1 (0.00036), CJ-2 (0.0012) and Gobi (0.0018) reference samples collected from the arid areas of North China. These results indicate that alkaline dust particles can adsorb gaseous MCAs in the atmosphere and largely control the long-range transport of LMW monocarboxylic acids from the Asian Continent to the western North Pacific Rim. Based on a good correlation

between monocarboxylic acids and nss-$Ca^{2+}$, it is very likely that organic acids in aerosols exist in the form of salts such as $Ca(HCOO)_2$, $Ca(HCOO)(CH_3COO)$ and/or $Ca(CH_3COO)_2$.

Prince et al. (2007) reported that gas phase acetic acid is adsorbed on the surface of calcite ($CaCO_3$), a major mineral of dust particles. Acetic acid can form calcium acetate in the atmosphere (Alexander et al., 2015). Vapor pressures of those organic anions are significantly lower than those of free monocarboxylic acids. Therefore, the acidity/alkalinity of aerosol surface is

an important factor to control the uptake of gaseous organic acids and thus organic acid salts can be long-range transported as particles in the atmosphere from the Asian Continent to the Japanese Islands without serious photochemical degradation. Zhang et al. (2012) reported that pH of wet deposition for the last two decades showed a slight increase in the southeast Tibetan Plateau, China due to the presence of $Ca^{2+}$ that is derived from Asian dust. We suggest that long-range atmospheric transport of LMW monocarboxylic acids associated with Asian dust over the Japanese Islands would be changed in the

future due to the changes in the emission of Asian dusts from the Asian Continent that are associated with global warming and changes in land use (Zhang et al., 2003; Song et al., 2016).

### 4.3 Major contributions of anthropogenic monocarboxylic acids

Benzoic acid is directly emitted from fossil fuel combustion (Kawamura et al., 1985) and also produced in the atmosphere by photo-oxidation of aromatic hydrocarbons such as toluene (Ho et al., 2015). Benzoic acid positively correlates with nss-$Ca^{2+}$

($r^2 = 0.81$). In addition, the average benzoic acid/nss-$Ca^{2+}$ ratio obtained for the Murodo-Daira snowpit samples (0.0029) is 3-4 orders of magnitude higher than those obtained from the Kosa reference materials such as CJ-1 (0.0000024), CJ-2 (0.0000033) and Gobi (0.0000078). Benzoic acid may be also adsorbed on the pre-existing particles via atmospheric titration of alkaline dust particles derived from the Asian Continent. The air mass trajectories arriving at the Murodo-Daira site have passed over North China where many industrial regions and mega-cities (e.g., Beijing) are located (Fig. 3). Therefore,

benzoic acid can be used as an anthropogenic tracer.

Formic plus acetic acids showed a strong positive correlation with benzoic acid ($r^2 = 0.81$), indicating that they are derived from anthropogenic sources in the Asian Continent. In contrast, nss-$K^+$, a tracer of biomass burning (Zhu et al., 2015), and nss-$F^-$, a tracer of coal-burning (Wang et al., 2005), did not show a positive correlation with formic plus acetic acids ($r^2 <$



0.03), indicating that biomass and coal burning is not a major source of monocarboxylic acids in the snow pit samples collected from the Murodo-Daira site near Mt. Tateyama.We consider that formic and acetic acids are both derived from anthropogenic and photochemical processes in the atmosphere of North China. They are adsorbed on the pre-existing particles via atmospheric titration with alkaline Kosa particles during the long-range atmospheric transport over the Japanese Islands.

The mean concentrations of formic and acetic acids in our samples in 2009 are higher than those reported in mountain snow samples from southern California (Kawamura et al., 1996), Tateyama (Kawamura et al., 2012) and south French Alps (Maupetit and Delmas, 1994) and ice core samples from Antarctica (de Angelis et al., 2012). Total MCA-C/DOC ratio (av. 21%) in 2009 is significantly higher than those reported in rainwater from Los Angeles (4.4%) (Kawamura et al., 2001), Shenzen, China (2.3%) (Huang et al., 2010), and reference dust materials (CJ-1: 2.0%, CJ-2: 2.9%, and Gobi: 3.3%). These results indicate that water-soluble LMW monocarboxylic acids in the snow pit samples near Mt. Tateyama constitute a significant fraction of water-soluble organic carbon, suggesting that entrainment of organic acids in snow flakes is significant during the atmospheric transport from China to Japan.

## 4.4 Minor contributions of biogenic monocarboxylic acids

Branched chain ($iC_4$-$iC_6$) monocarboxylic acids are produced by bacterial activity of *Bacteroides ruminicola*, *Megasphaera elsdenii*, and *Streptomyces avermitilis* (e.g., Allison, 1978; Hafner et al., 1991). It is of interest to note that $iC_5$ has not been reported in motor exhaust (Kawamura et al., 2000) and urban rainwater (Kawamura et al., 1996). However, it was detected in our snow pit samples. Lactic acid bacteria (*lactobacillus*) and plant tissues are known to produce lactic acid (Cabredo et al., 2009; Baker and El Saifi, 1953). *Lactobacillus* mainly exists in soil (Huysman and Verstraete, 1993). We found a strong positive correlation between branched chain ($iC_4$-$iC_6$) acids and lactic acid ($r^2 = 0.97$). Although the pathways of microbial production of branched chain monocarboxylic acids and lactic acid may be different, this strong correlation indicates that these organic acids are closely linked in the biosynthetic processes associated with bacterial activity in soils.

Branched chain ($iC_4$-$iC_6$) acids and lactic acid showed a positive correlation with nss-$Ca^{2+}$ ($r^2 = 0.72$). Maki et al. (2011, 2014) reported that bacterial communities are present in the layers of snow pit sequences at Murodo-Daira near the summit of Mt. Tateyama and are considered to be associated with Asian dust events. Although bacteria species responsible to branched monocarboxylic and lactic acids have not been reported in the Tateyama snow samples (Maki et al., 2014), our results suggest that branched chain monocarboxylic acids are produced by bacterial process in soils of the Asian Continent and transported over the Japanese Islands with Asian dust. However, contribution of biogenic monocarboxylic acids is much lower than anthropogenic monocarboxylic acids.



## 5 Summary and Conclusions

Low molecular weight normal ($C_1$-$C_{10}$), branched chain ($iC_4$-$iC_6$), hydroxyl (lactic and glycolic), and aromatic (benzoic and toluic isomers) monocarboxylic acids were detected in the snow pit samples collected from Murodo-Daira snowfield near the summit of Mt. Tateyama, central Japan. Acetic acid was detected as the dominant species (125 ng g$^{-1}$), followed by formic acid (85.7 ng g$^{-1}$) and isopentanoic acid (20.0 ng g$^{-1}$). Enhanced concentrations of monocarboxylic acids and nss-$Ca^{2+}$ were obtained in the snow pit samples with dust layers. We found that abundances of formic and acetic acids largely depend on non-sea-salt $Ca^{2+}$ ($r^2 = 0.73$). These acids positively correlated with benzoic acid ($r^2 = 0.81$) that is primarily produced by fossil fuel combustion and secondary photochemical oxidation of anthropogenic toluene, indicating that monocarboxylic acids were mainly of anthropogenic and photochemical origin. Formic plus acetic acids exponentially correlated with pH ($r^2 = 0.76$) (pH = 4.7-6.9). Alkaline dust particles may be subjected to atmospheric titration by gaseous monocarboxylic acids.

In addition, we analyzed reference dust materials including Chinese loess samples from the Tengger and Gobi deserts for the measurements of LMW monocarboxylic acids and inorganic ions. The ratio of total monocarboxylic acid/nss-$Ca^{2+}$ at the Murodo-Daira snowpit samples (0.27) was found to be significantly (two to three orders of magnitude) higher than those of Chinese loss reference samples (0.00036-0.0018). These comparisons suggest that gas phase monocarboxylic acids are easily adsorbed on the surface of pre-existing dust particles derived from the Asian Continent to result in organic acid salts. Our study demonstrates that Asian dust is a key factor to promote the long-range atmospheric transport of LMW monocarboxylic acids emitted and produced over North China to the western North Pacific Rim under a strong influence of the East Asian winter Monsoon. By forming the organic acid salts, LMW monocarboxylic acids can be more stabilized against the photochemical decomposition during long-range atmospheric transport.

## Acknowledgements

This study was in part supported by the Japan Society for the Promotion of Science (Grant-in-Aid Nos. 1920405 and 24221001). We thank E. Tachibana for the support of ion and dissolved organic carbon measurements and the students and researchers of University of Toyama for their help during the snow sampling. We also appreciate the helpful discussion with P. Q. Fu. The data of this paper are available upon request to K. Kawamura (kkawamura@isc.chubu.ac.jp).

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



Table 1. Descriptions of snowpack samples collected from a pit at Murodo-Daira near Mt. Tateyama, Japan in 2009 and 2011. Snowpack sample #4' was collected from different snow pit sequence parallel to sample #4.

| Year | Sample ID | Snow depth (cm) | Description |
|---|---|---|---|
| 2008-2009 | #1 | 325-335 | Weak dust layer. Asian dust event were observed on February 2, 2009 by a lidar over Toyama. Air masses are derived from the Taklamakan and Gobi desert. |
| | #2 | 410-420 | Clean snow layer |
| | #3 | 425-435 | Dusty snow layer. Asian dust event were observed on January 1, 2009 by a lidar over Toyama. Air masses are derived from the Taklamakan and Gobi desert. |
| | #4 | 520-530 | Dusty snow layer. Asian dust event were observed on December 10, 2008 by a lidar over Toyama. Air masses are derived from the Taklamakan and Gobi desert. |
| | #4' | 520-530 | Dusty snow layer |
| | #5 | 530-540 | Granular snow |
| 2010-2011 | #6 | 115-125 | Granular snow with ice plate |
| | #7 | 169-178 | Dusty and granular snow. Asian dust event were observed on February 22-24, 2011 by a lidar over Toyama. Air masses are derived from the Taklamakan and Gobi desert. |
| | #8 | 290-300 | Compacted snow layer |
| | #9 | 390-400 | Compacted snow layer |
| | #10 | 400-410 | Dusty and compacted snow. Asian dust event were observed on December 31, 2010 by a lidar over Toyama. Air masses are derived from the Taklamakan and Gobi desert. |
| | #11 | 430-440 | Compacted snow layer |
| | #12 | 460-466 | Dusty and compacted snow. Asian dust event were observed on December 25-26, 2010 by a lidar over Toyama. Air masses are derived from the Taklamakan and Gobi desert. |
| | #13 | 507-527 | Compacted snow with ice plate |
| | #14 | 542-548 | Dusty and compacted snow. Asian dust event were observed on December 6, 2010 by a lidar over Toyama. Air masses are derived from the Taklamakan and Gobi desert. |
| | #15 | 590-605 | Granular and compacted snow |
| | #16 | 630-635 | Granular snow |



Table 2. Concentrations (ng g$^{-1}$) of monocarboxylic acids in snowpack samples collected from a pit at Murodo-Daira near Mt. Tateyama, Japan in 2009 and 2011.

| Acid species | Snow sample ID (2009) | | | | | | Snow sample ID (2011) | | | | | | | | | | | Reference dust materials | | |
|---|---|---|---|---|---|---|---|---|---|---|---|---|---|---|---|---|---|---|---|---|
| | #1 | #2 | #3 | #4 | #4' | #5 | #6 | #7 | #8 | #9 | #10 | #11 | #12 | #13 | #14 | #15 | #16 | CJ-1 | CJ-2 | Gobi |
| **Aliphatic acids** | | | | | | | | | | | | | | | | | | | | |
| Formic, C$_1$ | 476 | 137 | 344 | 99.4 | 112 | 41.8 | 2.21 | 21.3 | 8.05 | 6.38 | 34.4 | 16.1 | 62.0 | 5.41 | 55.4 | 15.1 | 19.7 | 2420 | 3940 | 4402 |
| Acetic, C$_2$ | 708 | 273 | 456 | 121 | 140 | 51.2 | 9.01 | 52.6 | 31.0 | 25.1 | 61.5 | 36.4 | 50.8 | 21.9 | 40.1 | 22.2 | 25.6 | 1435 | 18540 | 11170 |
| Propionic, C$_3$ | 66.9 | 14.2 | 37.1 | 5.48 | 6.57 | 2.66 | 1.64 | 8.57 | 3.21 | 2.28 | 8.52 | 6.04 | 7.30 | 3.71 | 4.78 | 3.95 | 1.77 | 95 | 770 | 98 |
| Isobuthyric, iC$_4$ | 5.09 | 2.37 | 3.17 | 0.90 | 1.08 | 0.51 | 0.36 | 1.03 | 0.11 | 0.10 | 1.35 | 0.80 | 1.15 | 0.56 | 0.79 | 0.69 | 0.35 | n.d. | n.d. | 17 |
| Butyric, C$_4$ | 10.0 | 4.21 | 7.20 | 1.31 | 1.64 | 0.85 | 0.60 | 1.12 | 0.39 | 0.32 | 2.60 | 1.52 | 1.76 | 1.20 | 1.46 | 1.31 | 0.35 | 39 | 319 | 44 |
| Isopentanoic, iC$_5$ | 40.3 | 37.4 | 114 | 44.9 | 53.4 | 30.6 | 0.55 | 2.02 | 1.34 | 1.37 | 2.56 | 1.22 | 3.66 | 1.75 | 2.03 | 0.68 | 1.33 | 3 | 581 | 271 |
| Pentanoic, C$_5$ | 2.55 | 1.51 | 2.41 | 0.92 | 1.12 | 0.48 | 0.33 | 0.71 | 0.09 | 0.08 | 1.04 | 0.44 | 0.58 | 0.47 | 0.57 | 0.52 | 0.17 | 15 | 204 | 25 |
| Isohexanoic, iC$_6$ | n.d. | n.d. | n.d. | n.d. | n.d. | 0.12 | 0.03 | 0.04 | 0.01 | n.d. | 0.08 | 0.01 | 0.03 | 0.02 | 0.06 | 0.09 | n.d. | n.d. | n.d. | n.d. |
| Hexanoic, C$_6$ | 4.03 | 3.70 | 4.38 | 4.23 | 5.19 | 1.46 | 0.76 | 0.93 | 0.09 | 0.04 | 1.59 | 0.50 | 0.83 | 0.60 | 1.37 | 0.74 | 0.58 | 5 | 227 | 39 |
| Heptanoic, C$_7$ | 2.01 | 1.03 | 2.09 | 0.61 | 0.73 | 0.74 | 0.03 | 0.41 | 0.11 | 0.08 | 1.39 | 0.11 | 0.49 | 0.25 | 0.29 | 0.37 | 0.13 | 3 | 82 | 0 |
| Octanoic, C$_8$ | 1.04 | 0.25 | 1.38 | 1.03 | 1.26 | 0.55 | 0.16 | 0.15 | 0.02 | 0.01 | 0.46 | 0.11 | 0.09 | 0.15 | 0.23 | 0.19 | 0.07 | 4 | 125 | 14 |
| Nonanoic, C$_9$ | 6.47 | 7.23 | 5.50 | 3.66 | 4.78 | 3.61 | 1.12 | 1.38 | 0.14 | 0.04 | 1.62 | 0.73 | 0.88 | 1.08 | 1.09 | 0.79 | 0.66 | 47 | 1400 | 3255 |
| Decanoic, C$_{10}$ | 3.57 | 0.38 | 2.68 | 2.40 | 2.88 | 1.40 | 0.14 | 0.36 | 0.05 | 0.38 | 0.69 | 0.32 | 0.32 | 0.31 | 0.45 | 0.42 | 0.25 | n.d. | n.d. | n.d. |
| Sub total | 1330 | 481 | 981 | 286 | 331 | 136 | 16.9 | 90.6 | 44.7 | 36.1 | 117.8 | 64.3 | 129.9 | 37.4 | 108.6 | 50.4 | 51.0 | 4066 | 26190 | 19340 |
| **Aromatic acids** | | | | | | | | | | | | | | | | | | | | |
| Benzoic, Benz | 6.89 | 3.75 | 8.74 | 2.02 | 2.28 | 1.29 | 0.25 | 1.14 | 0.12 | 0.08 | 3.47 | 0.61 | 1.00 | 0.93 | 1.98 | 1.12 | 0.21 | 26 | 62 | 68 |
| o-toluic | n.d. | n.d. | 0.04 | 0.07 | 0.06 | n.d. | n.d. | 0.01 | n.d. | n.d. | 0.02 | n.d. | n.d. | 0.01 | 0.01 | 0.01 | n.d. | n.d. | n.d. | n.d. |
| m-toluic | 0.44 | 0.71 | 0.50 | 0.33 | 0.37 | 0.30 | 0.03 | 0.08 | n.d. | n.d. | 0.05 | n.d. | 0.04 | 0.02 | 0.02 | 0.02 | n.d. | n.d. | n.d. | n.d. |
| p-toluic | 0.09 | 0.06 | 0.11 | n.d. | 0.03 | 0.03 | 0.01 | 0.02 | 0.00 | n.d. | 0.07 | 0.01 | 0.03 | 0.02 | 0.03 | 0.02 | 0.01 | n.d. | n.d. | n.d. |
| Sub total | 7.42 | 4.53 | 9.39 | 2.42 | 2.74 | 1.62 | 0.29 | 1.25 | 0.12 | 0.08 | 3.61 | 0.62 | 1.06 | 0.97 | 2.04 | 1.16 | 0.22 | 26 | 62 | 68 |
| **Hydroxyacids** | | | | | | | | | | | | | | | | | | | | |
| Lactic, Lac | 1.46 | 1.11 | 5.06 | 1.73 | 1.89 | 1.26 | 0.16 | 0.01 | 0.15 | 0.28 | 0.07 | 0.15 | 0.21 | 0.21 | 0.27 | 0.38 | 0.14 | 192 | 2124 | 1215 |
| Glycolic, Glyco | 0.08 | 0.19 | 0.70 | 0.28 | 0.33 | 0.16 | 0.04 | 0.01 | 0.15 | 0.19 | 0.05 | 0.12 | 0.21 | 0.31 | 0.30 | 0.32 | 0.20 | 112 | 1020 | 385 |
| Sub total | 1.55 | 1.30 | 5.76 | 2.02 | 2.22 | 1.42 | 0.20 | 0.02 | 0.30 | 0.47 | 0.12 | 0.26 | 0.42 | 0.52 | 0.56 | 0.70 | 0.34 | 304 | 3144 | 1600 |
| DOC | 1360 | 508 | 2380 | 865 | 936 | 469 | 507 | 904 | 544 | 381 | 1580 | 723 | 427 | 743 | 704 | 2110 | 576 | 73000 | 403000 | 267000 |
| Total MA-C/DOC(%) | 35.6 | 36.7 | 15.8 | 13.1 | 14.1 | 12.1 | 1.5 | 3.9 | 3.2 | 3.7 | 2.9 | 3.5 | 10.8 | 2.1 | 5.4 | 1.0 | 3.2 | 2.0 | 2.9 | 3.3 |



Table 3. Concentrations (ng g$^{-1}$) of major ions and pH in snowpack samples collected from a pit at Murodo-Daira near Mt. Tateyama, Japan in 2009 and 2011.

| Inorganic species | Snow sample ID (2009) | | | | | | Snow sample ID (2011) | | | | | | | | | | | Reference dust materials | | |
|---|---|---|---|---|---|---|---|---|---|---|---|---|---|---|---|---|---|---|---|---|
| | #1 | #2 | #3 | #4 | #4' | #5 | #6 | #7 | #8 | #9 | #10 | #11 | #12 | #13 | #14 | #15 | #16 | CJ-1 | CJ-2 | Gobi |
| Anion | | | | | | | | | | | | | | | | | | | | |
| F$^-$ | 96 | 15 | 115 | 3 | 42 | 20 | 19 | 27 | 16 | 15 | 94 | 20 | 10 | 17 | 27 | 21 | 10 | 374000 | 148000 | 43600 |
| MSA$^-$ | 1080 | 62 | 1250 | 204 | 172 | 83 | 83 | 62 | 64 | 75 | 128 | 93 | 51 | 129 | 117 | 66 | 74 | 665000 | 224000 | 215000 |
| NO$_3^-$ | 534 | 130 | 458 | 316 | 224 | 150 | 791 | 1340 | 208 | 114 | 2020 | 88 | 104 | 428 | 1120 | 843 | 174 | 126000 | 1376000 | 138000 |
| SO$_4^{2-}$ | 1250 | 430 | 1460 | 728 | 536 | 364 | 845 | 1360 | 439 | 315 | 3010 | 310 | 260 | 549 | 1330 | 1070 | 282 | 2493000 | 24038000 | 1853000 |
| Total | 2960 | 637 | 3280 | 1250 | 974 | 617 | 1740 | 2790 | 727 | 520 | 5250 | 511 | 424 | 1120 | 2600 | 2000 | 540 | 3659000 | 25806000 | 2251000 |
| nss-F$^-$ | 96 | 15 | 115 | 3 | 42 | 19 | 19 | 27 | 16 | 15 | 94 | 20 | 9 | 17 | 27 | 21 | 10 | 374000 | 148000 | 43400 |
| nss-SO$_4^{2-}$ | 434 | 112 | 155 | 325 | 120 | 98 | 756 | 903 | 293 | 220 | 2440 | 215 | n.d. | 453 | 1040 | 967 | 192 | 2008000 | 23010000 | 1592000 |
| Cation | | | | | | | | | | | | | | | | | | | | |
| Na$^+$ | 3240 | 1270 | 5210 | 1610 | 1660 | 1060 | 356 | 1840 | 586 | 380 | 2310 | 380 | 1420 | 385 | 1160 | 417 | 362 | 1942000 | 4111000 | 1047000 |
| NH$_4^+$ | 111 | 41 | 200 | 268 | 243 | 78 | 235 | 291 | 54 | 36 | 842 | 47 | 34 | 56 | 517 | 190 | 41 | 336000 | 1460000 | 18700 |
| K$^+$ | 215 | 16 | 292 | 148 | 105 | 96 | 86 | 111 | n.d. | 50 | 302 | n.d. | 40 | n.d. | 119 | 43 | n.d. | 943000 | 4614000 | 2148000 |
| Ca$^{2+}$ | 3120 | 485 | 3390 | 1600 | 1890 | 505 | 184 | 639 | 148 | 140 | 1060 | 113 | 515 | n.d. | 574 | 220 | 200 | 10798000 | 18877000 | 8864000 |
| Mg$^{2+}$ | 190 | 6 | 195 | 334 | 152 | 24 | 35 | 127 | n.d. | 13 | 78 | n.d. | 19 | n.d. | 35 | 33 | n.d. | 1869000 | 1045000 | 754000 |
| Total | 6880 | 1817 | 9290 | 3960 | 4050 | 1760 | 896 | 3010 | 787 | 619 | 4590 | 540 | 2020 | 442 | 2410 | 903 | 602 | 15908000 | 30107000 | 12831000 |
| nss-K$^+$ | 95 | n.d. | 99 | 88 | 44 | 57 | 72 | 43 | n.d. | 36 | 217 | n.d. | n.d. | n.d. | 76 | 28 | n.d. | 871000 | 4462000 | 2110000 |
| nss-Ca$^{2+}$ | 3000 | 436 | 3190 | 1540 | 1820 | 464 | 170 | 569 | 125 | 125 | 976 | 99 | 462 | n.d. | 530 | 204 | 186 | 10725000 | 18721000 | 8824000 |
| nss-Mg$^{2+}$ | n.d. | n.d. | n.d. | 140 | 152 | n.d. | n.d. | n.d. | n.d. | n.d. | n.d. | n.d. | n.d. | n.d. | n.d. | n.d. | n.d. | 1636000 | 552000 | 628000 |
| pH | 6.9 | 6.1 | 6.7 | 6.7 | 6.3 | 6.0 | 4.7 | 6.0 | 5.2 | 5.0 | 6.2 | 5.1 | 6.2 | 4.9 | 5.9 | 4.4 | 5.4 | - | - | - |



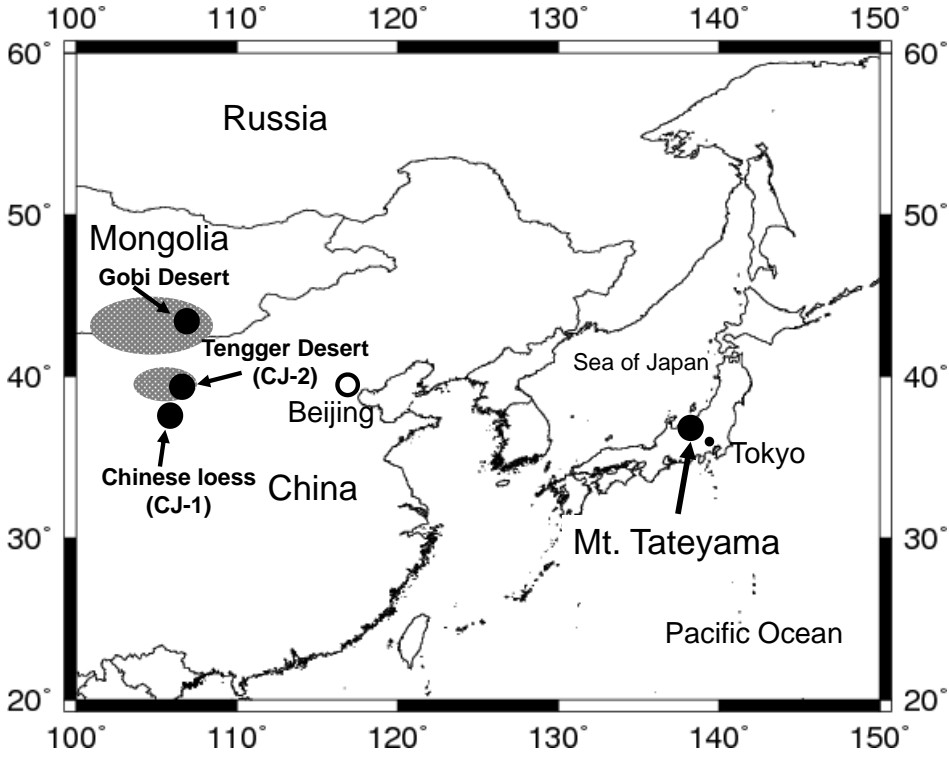

**Figure 1: Location of the snowpack sampling site (Murodo-Daira) near Mt. Tateyama, central Japan. Sites are also shown for the loess deposit reference samples, which were collected from Tengger and Gobi deserts in China and Mongolia (Nishikawa et al., 2000, 2013).**

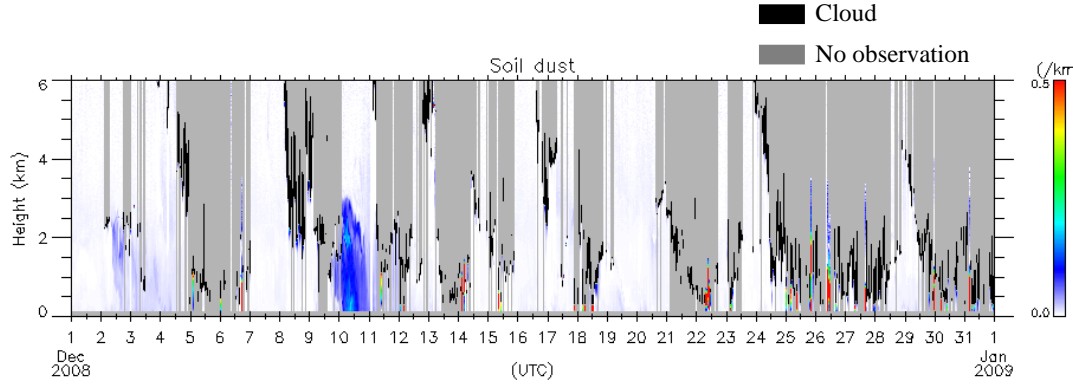

**Figure 2: Example of lidar measurements of dusts (extinction coefficient with color) and clouds (black) obtained at Imizu, Toyama (ca. 40 km northwest of Mt. Tateyama) during December 1-31, 2008. Gray shade means no data of dusts for the upper layers above clouds.**



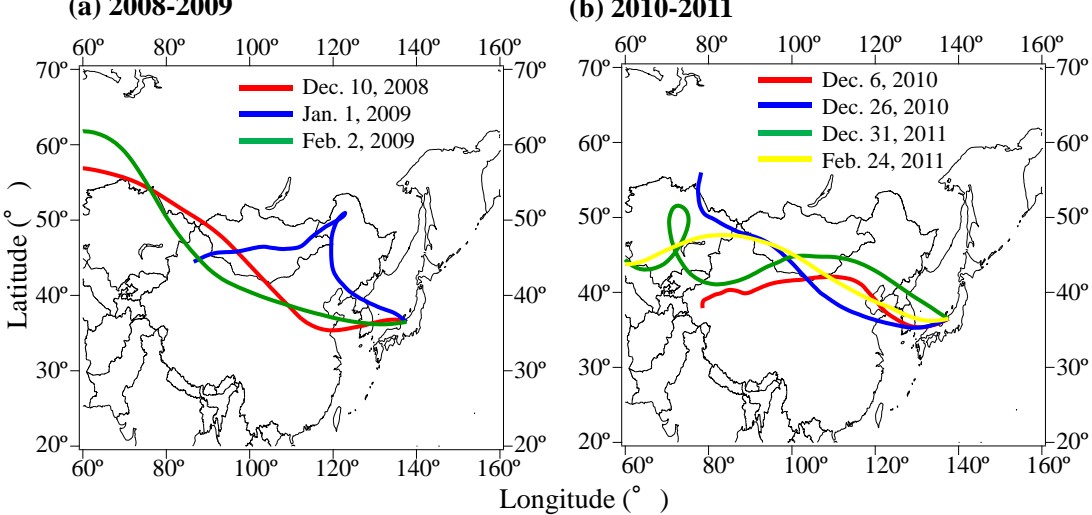

**Figure 3: Seven-day airmass back trajectories at a level of 3000 m a.s.l. over Murodo-Daira site in (a) 2008-2009 and (b) 2010-2011. Lines indicate trajectories the snow pit samples with dust layers.**

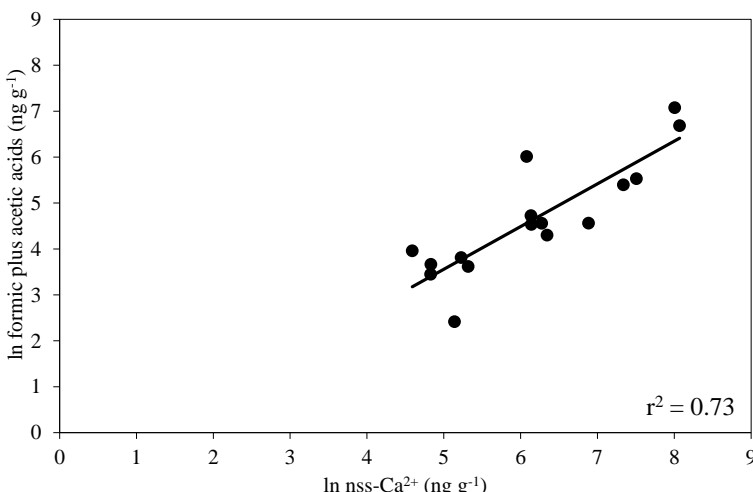

**Figure 4: Logarithm plots for the concentrations of formic plus acetic acids and nss-Ca²⁺ in the snow pit samples from Murodo-Daira site near Mt. Tateyama.**





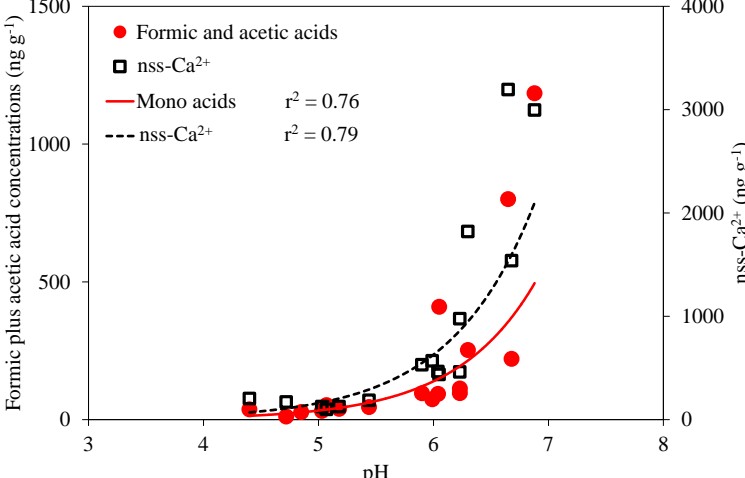

**Figure5: Correlation plots of formic plus acetic acids and nss-Ca$^{2+}$ in the snow pit samples against pH of the melt snow samples.**