# Peer review of "Long-range atmospheric transport of volatile monocarboxylic acids with Asian dust over high mountain snow site, central Japan"

_Atmospheric Chemistry and Physics, 2016_

## Referee Comment (RC1) · Anonymous Referee #1 · 5 Sep 2016

General Evaluation

This manuscript entitled "Long-range atmospheric transport of volatile monocarboxylic acids with Asian dust over high mountain snow site, central Japan" by T. Mochizuki et al. describes analytical results of monocarboxylic acids together with inorganic ions etc. in snowpack samples collected from a snow pit in Japan. The authors insist that Asian dusts can uptake volatile monocarboxylic acids during long-range transport and the dusts coated with organic acids act as effective ice nuclei to cause snowfalls. There have been only few reports which discuss the atmospheric uptake of organic compounds to Asian dust particles. Thus, the topic of this paper is interesting and also needs to be investigated from the view of the climate science. Hence, this manuscript is

recommended for publication in Atmospheric Chemistry and Physics, although needs several modifications.

Specific Comments

1) P.2, L23: "IN" has already been defined in the line 6.

2) p.3, L.30: "The sample was concentrated down to 10 ml using a rotary evaporator under vacuum at 50 °C"; p.4, L.4: "... and then dried using a rotary evaporator under vacuum, followed by blown-down with pure nitrogen gas"

The authors should describe how long it usually takes to reduce the water volume via each process. Such information would help readers who will try this method.

3) p.5, L.24: "We found that differences in the concentrations of each monocarboxylic acids between sample #4 and sample #4' are within the analytical uncertainties."

How large was the analytical uncertainty? The QA/QC for all the analytical methods should be mentioned in the paper.

4) p.7, L.5: "...the ratios of Mg/Ca at the Murodo-Daira site ..."

Are the ratios based on the mass concentration or the molar concentration? Should be clearly described.

5) P.7, L.10: "To investigate the effect of Asian dust on LMW monocarboxylic acids, we plotted major LMW monocarboxylic acids (i.e., formic plus acetic acids) against nss-Ca2+ using all the data points (Fig. 4)."

Why did the authors use the log-log plot? Should be explained. In addition, p values for r2 should be added. Why did the author use r2 (coefficient of determination) instead of r (correlation coefficient)?

6) P.7, L.26: "Figure 5 presents the relationship between formic plus acetic acids and pH of melt snow"

What function did the authors use? In addition, p values for r2 should be described.

7) p.7, L.13: "The atmospheric transport of Asian dust over the Japanese Islands is a dominant factor to control the concentrations of nss-Ca2+ and formic and acetic acids in snow precipitations during winter to spring"

As the authors mentioned, it is reasonable that Asian dust is a dominant factor to control the [nss-Ca2+]. However, why is it also acceptable for the concentrations of formic and acetic acids?

8) p.7, L.21: "Average concentrations of formic and acetic acids and nss-Ca2+ in 2009 are higher than those in 2011. Total rainfall during December to February in Urumqi near the Taklamakan desert, China in 2009 (31 mm) was half of that in 2011 (60 mm), suggesting that soil moisture in 2009 should have been lower than that in 2011 and thus the soil surfaces in 2009 should have been more dried. The higher concentrations of organic acids and nss-Ca2+ in the 2009 snow pit samples should be caused by a strong influence of the Asian dust events in this year when the soil surfaces were more dried in the arid regions, although the detailed records of the Asian dust events in North China are not available."

This part seems too speculative. Does the water content on the dust surface really depend on the amount of rainfall? In addition, I wonder why the higher concentrations of the organic acids and Ca2+ should be caused by the more dried soil surfaces.

9) P.8., L.1: "The adsorption process of acidic components by aerosols may be different between gaseous organic acids and inorganic gases such as SO2 and NOx"

How different are they? Do the authors have any ideas?

10) P.8, L.9: "Based on a good correlation between monocarboxylic acids and nss-Ca2+, it is very likely that organic acids in aerosols exist in the form of salts such as Ca(HCOO)2, Ca(HCOO)(CH3COO) and/or Ca(CH3COO)2"

Did the authors check the ion balance?

11) P.8, L.15: "and thus organic acid salts can be long-range transported as particles in the atmosphere from the Asian Continent to the Japanese Islands without serious photochemical degradation"

How can the authors find "without serious photochemical degradation"?

12) P.8., L.29: "Therefore, benzoic acid can be used as an anthropogenic tracer."

Why can benzoic acid be used as an anthropogenic tracer? It is unclear which is the specific reason why the benzoic acid can be used as the tracer.

13) P.8, L.31: "Formic plus acetic acids showed a strong positive correlation with benzoic acid ($r^2 = 0.81$)"

Which function did you use for the correlation analysis? The scatter plot should be displayed. (Also for the other correlation analysis results.)

14) p.10, L.14: I think the authors have incorrectly used "loss" instead of "loess".

---

## Referee Comment (RC2) · Anonymous Referee #2 · 8 Sep 2016

This study correlates high levels of low molecular weight, monocarboxylic acids (LMW-MCA) with Asian dust in the Japanese snow pack. Since the primary sources of LMW-MCA are not associated with Asian dust events, the conclusion is the organic acids adsorb onto the dust particles during transport. This changes the surface chemistry of the dust particle and therefore its efficiency as ice nuclei in clouds. The study indicates that while organic acids adsorb on dust particles, coating of dust particles by sulfate or nitrate is not as efficient. The study further shows that the uptake of formic acid and acetic acid largely depended on the amount of Ca in the dust and therefore does not apply to all dust types. This is a good study and I recommend publication with minor corrections and revisions.

[Figure]

I have concerns regarding the conclusion that coated dust particles make better ice nuclei and can therefore cause heavier snowfall. Many studies have shown that coating bare dust particles with organics can reduce the ice nucleating properties of bare mineral dust depending on the chemical composition the dust (Kulkarni et al 2014). Based on the complexity of ice nucleation properties of dust, the authors have not provided enough evidence to claim that these organic acid coated dust particles can alter the snowfall. This does not seem to be an important component of the study, but is an attempt for the authors to provide relevance context. It can be left out.

Minor Issues:

Abstract: Remove last line of the abstract as there is no evidence presented to support this statement.

Page 2, line 1 When referring to urban, forest, marine, and Arctic samples are these air samples or water/snow samples?

Page 2 line 20 This paragraph focuses on the importance of snow to the region and then discusses how changes in surface chemistry and hygroscopicity may improve ice nucleation of dust. See above comment. Previous studies have shown bare mineral dust to be very efficient IN and that coating may decrease ice nucleation properties, but increase water nucleation. This study provides no evidence either way and this section distracts from the point of the study.

Page 3 line 10 "several dirty layers were recognized by visual observation due to the occurrence of Asian dusts." This is an awkward statement. How was dust differentiated from a soot layer for example, or other industrial pollutants? How was the occurrence of Asian dust verified?

Page 3 line 13 "In order to evaluate the homogeneity of snow samples within the same snow horizon with dust layer, . . ." Not sure what is meant by this statement. Need clarification.

[Figure]

Page 3 line 25 "The data of inorganic . . .." In addition to back trajectory and lidar data, were mineral, or crustal elemental fractions in the snow contaminants compared with reference material for the different Asian dust regions to verify dust from specific regions?

Page 3 line 30 Why and how was the pH of the samples adjusted to 8.5 to 9.0? If this is described in Kawamura and Kaplan 1984, then including it here just raises questions and isn't informative. This is again stated on page 4 line 4 without explanation.

Page 7 line 25 "Although the alkalinity of snow pit samples can be affected. . .were slightly acidic." Not sure what the relevance of this statement is. I think it is the use of "although" that is throwing me off.

Some, but not all grammatical clean-up

Abstract Line 17 remove "being" before consistent.

Page 1 line 30, comma after Kawamura citation.

Page 2, line 8 have a variety of sources (insert of)

Page 5 line 7 insert "the" before laser.

Page 7 line 9 change has to was before "involved"

Check uses of "although" and "however", the authors use these two conjunctions are used a lot and not always appropriately.

Refs Kulkarni G., Sanders C., Zhang K., Liu X., and Zhao C., 'Ice nucleation of bare and sulfuric acid-coated mineral dust particles and implication for cloud properties (2014) J. of Geophys. Res. DOI 10.1002/2014JD021567

---

## Author Comment (AC1) · 27 Sep 2016

: t-mochizuki@u-shizuoka-ken.ac.jp

The comment was uploaded in the form of a supplement:
http://www.atmos-chem-phys-discuss.net/acp-2016-623/acp-2016-623-AC1-supplement.zip

---

## Author Response (AR1)

**Authors Responses to Reviewers**

**We appreciate the helpful comments made by reviewers.**

Below, we indicate in detail the revisions made to the manuscript.

Text form the original is shown in blue.

Changes in the revised manuscript are shown in red.

**Referee #1**

Comments to Author

General Evaluation

This manuscript entitled "Long-range atmospheric transport of volatile monocarboxylic acids with Asian dust over high mountain snow site, central Japan" by T. Mochizuki et al. describes analytical results of monocarboxylic acids together with inorganic ions etc. in snowpack samples collected from a snow pit in Japan. The authors insist that Asian dusts can uptake volatile monocarboxylic acids during long-range transport and the dusts coated with organic acids act as effective ice nuclei to cause snowfalls. There have been only few reports which discuss the atmospheric uptake of organic compounds to Asian dust particles. Thus, the topic of this paper is interesting and also needs to be investigated from the view of the climate science. Hence, this manuscript is recommended for publication in Atmospheric Chemistry and Physics, although needs several modifications.

Specific Comments

*1) P.2, L23: "IN" has already been defined in the line 6.*

**Response:**

We have removed the entire description of IN following the comments provided by reviewer 2.

*2) p.3, L.30: "The sample was concentrated down to 10 ml using a rotary evaporator under vacuum at 50 °C"; p.4, L.4: "... and then dried using a rotary evaporator under vacuum, followed by blown-down with pure nitrogen gas"*

*The authors should describe how long it usually takes to reduce the water volume via each process. Such information would help readers who will try this method.*

**Response:**

We did not measure how long it takes to reduce the sample water volume via rotary

evaporator under vacuum. We have modified the following descriptions in the manuscript.

The sentence "The sample was concentrated down to 10 ml using a rotary evaporator under vacuum at 50 °C." has been revised to "The sample was concentrated down to 10 ml using a rotary evaporator under vacuum (20 mmHg) at 50 °C." (Page 3, Lines 27-28).

The sentence "The pH of the sample was checked to be 8.5–9.0 and then dried using a rotary evaporator under vacuum, followed by blown-down with pure nitrogen gas." has been revised to "The pH of the sample was checked to be 8.5–9.0 and then dried using a rotary evaporator under vacuum (20 mmHg), followed by blown-down with pure nitrogen gas for 30 seconds. The former process general requires 15-20 min." (Page 3, Line 32 – Page 4, Line 2).

*3) p.5, L24: "We found that differences in the concentrations of each monocarboxylic acids between sample #4 and sample #4' are within the analytical uncertainties."*
*How large was the analytical uncertainty? The QA/QC for all the analytical methods should be mentioned in the paper.*

**Response:**

Based on the comment, we modified the following sentences.

The sentence "Analytical errors using authentic standards were within 12%." has been revised to "Analytical errors in the GC/FID analysis using authentic standards were within 2%. Relative standard deviations based on triplicate analysis of real samples were within 12%." (Page 4, Lines 19-20).

The sentence "We found that differences in the concentrations of each monocarboxylic acids between sample #4 and sample #4' are within the analytical uncertainties." has been revised to "We found that differences in the concentrations of each monocarboxylic acids between sample #4 and sample #4' are comparable to the relative standard deviations." (Page 5, Lines 24-25).

*4) p.7, L.5: "... the ratios of Mg/Ca at the Murodo-Daira site ..."*
*Are the ratios based on the mass concentration or the molar concentration? Should be clearly described.*

**Response:**

 Based on the comment, we modified the following sentences.

The sentence "In addition, the ratios of Mg/Ca at the Murodo-Daira site in 2009 and 2011 are 0.08 and 0.12, respectively." has been changed to "In addition, the mass concentration ratios of Mg/Ca at the Murodo-Daira site in 2009 and 2011 are 0.08 and 0.12, respectively." (Page 7, Lines 5-6).

*5) P.7, L.10: "To investigate the effect of Asian dust on LMW monocarboxylic acids, we plotted major LMW monocarboxylic acids (i.e., formic plus acetic acids) against nss-Ca$^{2+}$ using all the data points (Fig. 4)."*

*Why did the authors use the log-log plot? Should be explained. In addition, p values for r$^2$ should be added. Why did the author use r$^2$ (coefficient of determination) instead of r (correlation coefficient)?*

**Response:**

 Instead of the log-log plot, we plotted Fig. 4 with a regular style. Fig. 4 shows a strong correlation between concentrations of formic plus acetic acids and nss-Ca$^{2+}$. The "r$^2$" has been changed to "r" in this manuscript. Based on the suggestion, we have modified Figure 4 with modified caption.

[Figure]

Figure 4: Linear regression plots between concentrations of formic plus acetic acids and nss-Ca$^{2+}$ in Mt. Tateyama snow samples. (Page 19, Lines 6-7).

 In addition, the sentence; "We found that formic plus acetic acids strongly correlate with nss-Ca$^{2+}$ (r$^2$ = 0.73)." has been modified to "Concentrations of formic plus acetic

acids were found to increase linearly with that of nss-Ca$^{2+}$ (r = 0.85, *p* < 0.001)." (Page 7, Lines, 11-12).

*6) P.7, L.26: "Figure 5 presents the relationship between formic plus acetic acids and pH of melt snow"*

*What function did the authors use? In addition, p values for r$^2$ should be described.*

**Response:**

Based on the suggestion, the sentences; "We found that formic plus acetic acids positively correlated with pH (r$^2$ = 0.76). Their concentrations exponentially increased at pH range of > 6.0. Interestingly, concentrations of nss-Ca$^{2+}$ positively correlate with pH (r$^2$ = 0.79) (Fig. 5)." have been changed to "Concentrations of formic plus acetic acids were found to increase exponentially with pH (r = 0.87, *p* < 0.001). Interestingly, concentrations of nss-Ca$^{2+}$ were also found to increase exponentially with pH (r = 0.89, *p* < 0.001) (Fig. 5)." (Page 7, Lines 20-22).

We have modified the Y axis (left and right) and caption of Fig. 5 as follows.

[Figure]

Figure 5: Correlation plots of natural logarithm of formic plus acetic acids and nss-Ca$^{2+}$ in the snow pit samples against pH of the melt snow samples. (Page 20, Lines 1-3).

*7) p.7, L.13: "The atmospheric transport of Asian dust over the Japanese Islands is a dominant factor to control the concentrations of nss-Ca$^{2+}$ and formic and acetic acids in snow precipitations during winter to spring"*

*As the authors mentioned, it is reasonable that Asian dust is a dominant factor to*

*control the [nss-Ca$^{2+}$]. However, why is it also acceptable for the concentrations of formic and acetic acids?*

**Response:**

We have modified the following descriptions in the manuscript.

The sentences; "The atmospheric transport of Asian dust over the Japanese Islands is a dominant factor to control the concentrations of nss-Ca$^{2+}$ and formic and acetic acids in snow precipitations during winter to spring, when the Asian dust activity maximizes in North China." have been revised to "Asian dust particles may be a carrier of formic and acetic acids via acid/base reaction and forming carboxylate salts, when the Asian dust activity maximizes in North China." (Page 7, Lines 13-14).

*8) p.7, L.21: "Average concentrations of formic and acetic acids and nss-Ca$^{2+}$ in 2009 are higher than those in 2011. Total rainfall during December to February in Urumqi near the Taklamakan desert, China in 2009 (31 mm) was half of that in 2011 (60 mm), suggesting that soil moisture in 2009 should have been lower than that in 2011 and thus the soil surfaces in 2009 should have been more dried. The higher concentrations of organic acids and nss-Ca$^{2+}$ in the 2009 snow pit samples should be caused by a strong influence of the Asian dust events in this year when the soil surfaces were more dried in the arid regions, although the detailed records of the Asian dust events in North China are not available."*

*This part seems too speculative. Does the water content on the dust surface really depend on the amount of rainfall? In addition, I wonder why the higher concentrations of the organic acids and Ca$^{2+}$ should be caused by the more dried soil surfaces.*

**Response:**

Our hypothesis was speculative. Based on the suggestion, we have modified the following sentences as follows.

The following sentences were deleted. "Total rainfall during December to February in Urumqi near the Taklamakan desert, China in 2009 (31 mm) was half of that in 2011 (60 mm) (Japan Meteorological Agency website: http://www.data.jma.go.jp/gmd/cpd/monitor/mainstn/obslist.php), suggesting that soil moisture in 2009 should have been lower than that in 2011 and thus the soil surfaces in 2009 should have been more dried. The higher concentrations of organic acids and nss-Ca$^{2+}$ in the 2009 snow pit samples should be caused by a strong influence of the Asian dust events in this year when the soil surfaces were more dried in the arid regions, although the detailed records of the Asian dust events in North China are not available."

Instead we added the following sentences; "This may be related to a strong influence of the Asian dust events. However, the detailed records of the Asian dust events in North China are not available." (Page 7, Lines 16-18).

*9) P.8., L.1: "The adsorption process of acidic components by aerosols may be different between gaseous organic acids and inorganic gases such as $SO_2$ and $NO_x$"*
*How different are they? Do the authors have any ideas?*
**Response:**

Our previous study showed a positive correlation between water soluble organic nitrogen (WSON) and nss-$Ca^{2+}$ (Mochizuki et al., 2016). On the other hand, water soluble inorganic nitrogen (WSIN) ($NH_4^+$ plus $NO_3^-$) did not show a correlation with nss-$Ca^{2+}$ (Mochizuki et al., 2016). We think that Asian dust is a factor to control the long-range transport of organic acids, WSON, WSIN, and inorganic ions including $SO_4^{2-}$, but the production process of salts, adsorption process on dust particles, and removal process in the atmosphere may be different. These phenomena do not make clear understanding in our study.

We have deleted the following sentences in the revised MS because they are too speculative.

"In contrast, $SO_4^{2-}$ and $NO_3^-$ did not show any correlation with pH (r < 0.10). These results may suggest that, during the atmospheric transport, dust particles do not efficiently uptake inorganic acids that are mostly present as particles. The adsorption process of acidic components by aerosols may be different between gaseous organic acids and inorganic gases such as $SO_2$ and $NO_x$, which are precursors of $SO_4^{2-}$ and $NO_3^-$, respectively."

*10) P.8, L.9: "Based on a good correlation between monocarboxylic acids and nss-$Ca^{2+}$, it is very likely that organic acids in aerosols exist in the form of salts such as $Ca(HCOO)_2$, $Ca(HCOO)(CH_3COO)$ and/or $Ca(CH_3COO)_2$"*
*Did the authors check the ion balance?*
**Response:**

We calculated ion balance. The above-mentioned points were added in the revised MS.

"We calculated ion balance in the snow pit at the Murodo-Daira site near Mt. Tateyama. In this study, we could not use the data of chloride ion ($Cl^-$) because of the

addition of $HgCl_2$ into snow samples. To calculate ion balance, we used equivalent ratio of $Cl^-$ to $Na^+$ (1.26) obtained in the same snow pit in 2011 (Watanabe et al., 2012). Figure 6 shows total cations ($Na^+$, $NH_4^+$, $K^+$, $Ca^{2+}$, and $Mg^{2+}$) against total anions ($F^-$, $MSA^-$, $NO_3^-$, $SO_4^{2-}$ and organic anions including normal ($C_1$-$C_{10}$), branched chain ($iC_4$-$iC_6$), aromatic (benzoic and toluic acid isomers), and hydroxyl (lactic and glycolic) monocarboxylic acids) (r = 0.95, $p$ < 0.001). The slope (1.26) of more than unity indicates that excess cations exist in the snow pit at the Murodo-Daira site near Mt. Tateyama, although $CO_3^-$ and $HCO_3^-$, and unidentified organic anions were not determined." (Please see Page 7, Lines 26-32).

We have added Figure 6 with the caption as below.

[Figure]

Figure 6: Linear regression plots between total cation equivalents (neq) and total anion equivalents (neq) in melt snow samples at the Murodo-Daira site near Mt. Tateyama. (Page 20, Lines 5-7).

*11) P.8, L.15: "and thus organic acid salts can be long-range transported as particles in the atmosphere from the Asian Continent to the Japanese Islands without serious photochemical degradation"*

*How can the authors find "without serious photochemical degradation"?*

**Response:**

We have added the following sentences in response to the above comment.

"In addition, the lifetimes of formic and acetic acids with OH radical are estimated

to be 25 days and 10 days, respectively, at -13 °C assuming the OH concentration of $1.0 \times 10^6$ molecules cm$^{-3}$ (Paulot et al., 2011). This timescale is much longer than that of atmospheric transport time of air mass from the Asian Continent to Mt. Tateyama." (Page 8, Lines 11-14).

*12) P.8., L.29: "Therefore, benzoic acid can be used as an anthropogenic tracer."*
*Why can benzoic acid be used as an anthropogenic tracer? It is unclear which is the specific reason why the benzoic acid can be used as the tracer.*

**Response:**

We are sorry for the unclear description. We modified following sentences in the revised MS.

The following sentences were deleted; "Benzoic acid is directly emitted from fossil fuel combustion (Kawamura et al., 1985) and also produced in the atmosphere by photo-oxidation of aromatic hydrocarbons such as toluene (Ho et al., 2015). Benzoic acid positively correlates with nss-Ca$^{2+}$ ($r^2 = 0.81$). In addition, the average benzoic acid/nss-Ca$^{2+}$ ratio obtained for the Murodo-Daira snowpit samples (0.0029) is 3-4 orders of magnitude higher than those obtained from the Kosa reference materials such as CJ-1 (0.0000024), CJ-2 (0.0000033) and Gobi (0.0000078). Benzoic acid may be also adsorbed on the pre-existing particles via atmospheric titration of alkaline dust particles derived from the Asian Continent. The air mass trajectories arriving at the Murodo-Daira site have passed over North China where many industrial regions and mega-cities (e.g., Beijing) are located (Fig. 3). Therefore, benzoic acid can be used as an anthropogenic tracer."

Instead, we added the following sentences in the revised MS; "Benzoic acid is directly emitted from fossil fuel combustion (Kawamura et al., 1985) and also produced in the atmosphere by photo-oxidation of aromatic hydrocarbons such as toluene (Forstner et al., 1997), which are derived from human activities. Benzoic acid positively correlates with nss-Ca$^{2+}$ ($r = 0.90$, $p < 0.001$) (Fig. 7a). In addition, the average benzoic acid/nss-Ca$^{2+}$ ratio obtained for the Murodo-Daira snow pit samples (0.0029) is 3-4 orders of magnitude higher than those obtained from the Kosa reference materials such as CJ-1 (0.0000024), CJ-2 (0.0000033) and Gobi (0.0000078). Benzoic acid may also be adsorbed on the pre-existing particles via atmospheric titration of alkaline dust particles derived from the Asian Continent. The air mass trajectories arriving at the Murodo-Daira site have passed over North China where many industrial regions and mega-cities (e.g., Beijing) are located (Fig. 3)." (Please see Page 8, Lines 23-30).

We have deleted the following reference to the references section.

Ho, K. F., Huang, R. -J., Kawamura, K., Tachibana, E., Lee, S. C., Ho, S. S. H., Zhu, T., and Tian, L. W.: Dicarboxylic acids, ketocarboxylic acids, α-dicarbonyls, fatty acids and benzoic acid in PM2.5 aerosol collected during CAREBeijing-2007: an effect of traffic restriction on air quality, Atmos. Chem. Phys., 15, 3111–3123, 2015.

*13) P.8, L.31: "Formic plus acetic acids showed a strong positive correlation with benzoic acid ($r^2 = 0.81$)"*

*Which function did you use for the correlation analysis? The scatter plot should be displayed. (Also for the other correlation analysis results.)*

**Response:**

Based on the suggestion, we have added Figure 7 with the caption as below.

[Figure]

Figure 7: Linear regression plots between (a) concentrations of benzoic acid and nss-$Ca^{2+}$, (b) formic plus acetic acids and benzoic acid, (c) formic plus acetic acids and nss-$K^+$, and (d) formic plus acetic acids and nss-$F^-$ in Mt. Tateyama snow samples. (Page 21, Lines 1-4).

We have modified the following sentences.

The sentence "Benzoic acid positively correlates with nss-Ca$^{2+}$ (r$^2$ = 0.81)." has been revised to "Benzoic acid positively correlated with nss-Ca$^{2+}$ (r = 0.90, $p$ < 0.001). (Fig. 7a)." (Page 8, Line 25).

The phrase; "Formic plus acetic acids showed a strong positive correlation with benzoic acid (r$^2$ = 0.81), …" has been changed to "Formic plus acetic acids showed a strong positive correlation with benzoic acid (r = 0.90, $p$ < 0.001) (Fig. 7b), …" (Page 8, Line 31).

The sentences; "In contrast, nss-K$^+$, a tracer of biomass burning (Zhu et al., 2015), and nss-F$^-$, a tracer of coal-burning (Wang et al., 2005), did not show a positive correlation with formic plus acetic acids (r$^2$ < 0.03), indicating that biomass and coal …" have been modified to "In contrast, nss-K$^+$, a tracer of biomass burning (Zhu et al., 2015) did not show a positive correlation with formic plus acetic acids (r = 0.18, $p$ < 0.6) (Fig. 7c). nss-F$^-$, a tracer of coal-burning (Wang et al., 2005) shows a positive correlation with formic plus acetic acids (r = 0.72, $p$ < 0.01) (Fig. 7d), however, they were rather scattered. Biomass and coal …" (Page 8, Line 32 – Page 9, Line 2).

Based on the suggestion, we have added Figure 8.

[Figure]

Figure 8: Linear regression plots between concentrations of branched chain (iC$_4$-iC$_6$) monocarboxylic acids and lactic acid in Mt. Tateyama snow samples. (Page 21, Lines 6-8).

We have modified the following sentence.

The sentence; "We found a strong positive correlation between branched chain (iC$_4$-iC$_6$) acids and lactic acid (r$^2$ = 0.97)." has been revised to "We found a strong positive correlation between branched chain (iC$_4$-iC$_6$) acids and lactic acid (r = 0.98, $p$ < 0.001) (Fig. 8)." (Page 9, Lines 19-20).

Based on the suggestion, we have added Figure 9.

[Figure]

Figure 9: Linear regression plots between (a) concentrations of branched chain (iC$_4$-iC$_6$) monocarboxylic acids and nss-Ca$^{2+}$ and (b) lactic acid and nss-Ca$^{2+}$ in Mt. Tateyama snow samples. (Page 22, Lines 1-3).

We have modified the following sentence.

The sentence "Branched chain (iC$_4$-iC$_6$) acids and lactic acid showed a positive correlation with nss-Ca$^{2+}$ (r$^2$ = 0.72)." has been revised to "Branched chain (iC$_4$-iC$_6$) acids (r = 0.85, $p$ < 0.001) (Fig. 9a) and lactic acid (r = 0.81, $p$ < 0.001) (Fig. 9b) showed a positive correlation with nss-Ca$^{2+}$." (Page 9, Lines 22-23).

In addition, we have modified the following sentences.

The sentence "We found a strong correlation (r$^2$=0.73) between formic plus acetic acids and non-sea-salt Ca$^{2+}$ that is a proxy of Asian dust." has been revised to "We found a strong correlation (r = 0.85, $p$ < 0.001) between formic plus acetic acids and non-sea-salt Ca$^{2+}$ that is a proxy of Asian dust." (Page 1, Lines 15-16).

The phrase "Formic plus acetic acids also showed a positive correlation (r$^2$=0.81) with benzoic acid …" has been revised to "Formic plus acetic acids also showed a positive correlation (r = 0.90, $p$ < 0.001) with benzoic acid …" (Page 1, Lines 18-19).

The sentence "Entrainment of organic acids by dusts is supported by a good correlation ($r^2$=0.76) between formic plus acetic acids and pH of melt snow samples." has been revised to "Entrainment of organic acids by dusts is supported by a good correlation ($r = 0.86$, $p < 0.001$) between formic plus acetic acids and pH of melt snow samples." (Page 1, Lines 24-25).

The sentences "We found that abundances of formic and acetic acids largely depend on non-sea-salt $Ca^{2+}$ ($r^2 = 0.73$). These acids positively correlated with benzoic acid ($r^2 = 0.81$) that is primarily produced by fossil fuel combustion and secondary photochemical oxidation of anthropogenic toluene, indicating that monocarboxylic acids were mainly of anthropogenic and photochemical origin. Formic plus acetic acids exponentially correlated with pH ($r^2 = 0.76$) (pH = 4.7-6.9)." have been revised to "We found that abundances of formic and acetic acids largely depend on non-sea-salt $Ca^{2+}$ ($r = 0.85$, $p < 0.001$). These acids positively correlated with benzoic acid ($r = 0.90$, $p < 0.001$) that is primarily produced by fossil fuel combustion and secondary photochemical oxidation of anthropogenic toluene and other aromatic hydrocarbons, indicating that monocarboxylic acids were mainly of anthropogenic and photochemical origin. Formic plus acetic acids exponentially correlated with pH ($r = 0.86$, $p < 0.001$) (pH = 4.7-6.9)." (Page 10, Lines 3-7).

*14) p.10, L.14: I think the authors have incorrectly used "loss" instead of "loess".*
**Response:**
Corrected to "loess". (Page 10, Line 12).

In addition, we have modified the following sentences.
The sentence "Details of analytical procedure were described previously (Kawamura et al., 2012)." has been revised to "Details of analytical procedure were described previously except for the pH adjustment with KOH solution (Kawamura et al., 2012)." (Page 4, Lines 16-17).

The sentence "They are adsorbed on the pre-existing particles via atmospheric titration with alkaline Kosa particles during the long-range atmospheric transport over the Japanese Islands." has been revised to "They are adsorbed on the pre-existing

alkaline Kosa particles via atmospheric titration during the long-range atmospheric transport over the Japanese Islands." (Page 9, Lines 4-5).

The phrase "Total MCA-C/DOC ratio (av. 21%) in 2009 is significantly higher than those reported in rainwater from Los Angeles …" has been revised to "Total MCA-C/DOC ratio (av. 21%) in 2009 is significantly higher than those reported in rainwater samples from Los Angeles …" (Page 9, Lines 8-9).

The phrase "… suggesting that entrainment of organic acids in snow flakes is significant during the atmospheric transport from China to Japan." has been revised to "… suggesting that entrainment of organic acids in alkaline dusts and snow flakes is significant during the atmospheric transport from China to Japan." (Page 9, Lines 12-13).

We have added the following papers to the reference section.

Forstner, H. J. L., Flagan, R. C., and Seinfeld, J. H.: Secondary organic aerosol from the photooxidation of aromatic hydrocarbons: Molecular composition, Environ. Sci. Technol. 31, 1345-1358, 1997.

Watanabe, K., Nishimoto, D., Ishita, S., Eda, N., Uehara, Y., Takahashi, G., Kunori, N., Kawakami, T., Shimada, W., Aoki, K., and Kawada, K.: Formaldehyde and hydrogen peroxide concentrations in the snow cover at Murododaira, Mt. Tateyama Japan, Bull. Glaciol. Res., 30, 33-40, 2012.

**Authors Responses to Reviewers**

**We appreciate the helpful comments made by reviewers.**

Below, we indicate in detail the revisions made to the manuscript.

Text form the original is shown in blue.

Changes in the revised manuscript are shown in red.

**Referee #2**

Comments to the Author

This study correlates high levels of low molecular weight, monocarboxylic acids (LMWMCA) with Asian dust in the Japanese snow pack. Since the primary sources of LMWMCA are not associated with Asian dust events, the conclusion is the organic acids adsorb onto the dust particles during transport. This changes the surface chemistry of the dust particle and therefore its efficiency as ice nuclei in clouds. The study indicates that while organic acids adsorb on dust particles, coating of dust particles by sulfate or nitrate is not as efficient. The study further shows that the uptake of formic acid and acetic acid largely depended on the amount of Ca in the dust and therefore does not apply to all dust types. This is a good study and I recommend publication with minor corrections and revisions.

*I have concerns regarding the conclusion that coated dust particles make better ice nuclei and can therefore cause heavier snowfall. Many studies have shown that coating bare dust particles with organics can reduce the ice nucleating properties of bare mineral dust depending on the chemical composition the dust (Kulkarni et al 2014). Based on the complexity of ice nucleation properties of dust, the authors have not provided enough evidence to claim that these organic acid coated dust particles can alter the snowfall. This does not seem to be an important component of the study, but is an attempt for the authors to provide relevance context. It can be left out.*

**Response:**

Based on the comment 3), we have modified the following descriptions in the manuscript. Please see our responses to comment 3).

Minor Issues:

*1) Abstract: Remove last line of the abstract as there is no evidence presented to support this statement.*

**Response:**

As the referee suggested, we have rephrased the sentence "Our study demonstrates that Asian alkaline dusts can uptake volatile monocarboxylic acids during long-range transport, and the dusts coated with organic acids may act as effective ice nuclei to cause a heavy snowfall." to "Our study suggests that Asian alkaline dusts may be a carrier of volatile monocarboxylic acids." (Page 1, Lines 25-26).

*2) Page 2, line 1 When referring to urban, forest, marine, and Arctic samples are these air samples or water/snow samples?*

**Response:**

We are sorry for the unclear description. We have modified the following sentence.

The sentence "They have been reported in urban (Kawamura et al., 2000) forest (Andreae et al., 1988), high mountain (Preunkert et al., 2007), marine (Miyazaki et al., 2014), and Arctic samples (Talbot et al., 1992; Legrand et al., 2004)." has been revised to "Gaseous and particulate formic and acetic acids have been reported in urban (Kawamura et al., 2000), forest (Andreae et al., 1988), high mountain (Preunkert et al., 2007), marine (Miyazaki et al., 2014), and Arctic samples (Legrand et al., 2004)." (Page 1, Line 30 - Page 2, Line 2).

We have deleted the following reference to the references section.
Talbot, R. W., Vijgen, A. S., and Harris, R. C.: Soluble species in the Arctic summer troposphere: acidic gases, aerosols, and precipitation, J. Geophys. Res., 97, 16531-16543, 1992.

*3) Page 2 line 20 This paragraph focuses on the importance of snow to the region and then discusses how changes in surface chemistry and hygroscopicity may improve ice nucleation of dust. See above comment. Previous studies have shown bare mineral dust to be very efficient IN and that coating may decrease ice nucleation properties, but increase water nucleation. This study provides no evidence either way and this section distracts from the point of the study.*

**Response:**

Kulkarni et al. (2014) and references therein reported that ice nucleation potential of dust particles coated with organics are lower than that of bare dust particles. These are very important. However, our study do not clarify the relationship between ice nucleation potential and Asian dust particles at Murodo-Daira near the summit of Mt.

Tateyama. Based on the suggestion, we have modified the following descriptions in the manuscript.

The following sentences were deleted; "Because these snow precipitations are the sources for agricultural and drinking waters in those areas, snow precipitations in the high mountains are important for the water cycles in Japan. Ice nuclei (IN) supplied from the Asian Continent through long-range atmospheric transport and the water vapours supplied from the Sea of Japan during the winter Asian Monsoon are two important components to promote the heavy snow over the western part of Japan. Asian dust particles may act as effective IN to result in a heavy snow over the Japanese Alps. During the formation of snow crystals, chemical compositions of dust surface may be critical to control the hygroscopic properties of dust particles (Creamean et al., 2013)."

We have deleted the following reference in the references section.

Creamean, J. M., Suski, K. J., Rosenfeld, D., Cazorla, A., DeMott, P. J., Sullivan, R. C., White, A. B., Ralph, F. M., Minnis, P., Comstock, J. M., Tomlinson, J. M., and Prather, K. A.: Dust and biological aerosols from the Sahara and Asia influence precipitation in the Western U.S., Science, 339, 1572-1578, 2013.

*4) Page 3 line 10 "several dirty layers were recognized by visual observation due to the occurrence of Asian dusts." This is an awkward statement. How was dust differentiated from a soot layer for example, or other industrial pollutants? How was the occurrence of Asian dust verified?*

**Response:**

We are sorry for the unclear description. In the section 2 stage, we do not decide the dust, soot, or other industrial pollutants. The word "dust" has been replaced with "dirty" in section 2. We have modified the following sentences in response to the above comment.

The following sentences were deleted; "Table 1 provides descriptions of snow samples collected from the snow pit sequence, in which several dirty layers were recognized by visual observation due to the occurrence of Asian dusts. Five snowpack samples including three dust layers were collected from the pit sequence in April 18, 2009. Eleven snowpack samples including four dust layers were collected from the pit sequence in April 17, 2011. In order to evaluate the homogeneity of snow samples within the same snow horizon with dust layer, another snowpack sample (#4') was collected at ca. 1 m away from the location of sample #4 in 2009. Because the thickness

of dust layers in snow pit sequence is ca. 10 cm or more, dusts are deposited together with snowflakes during snow precipitation rather than dry deposition."

We added the following sentenced in the revised MS; "Table 1 provides descriptions of snow samples collected from the snow pit sequence, in which several brown-colored dirty layers were recognized by visual observation. Five snowpack samples including three dirty layers were collected from the pit sequence in April 18, 2009. Eleven snowpack samples including four dirty layers were collected from the pit sequence in April 17, 2011. In order to evaluate the consistent distribution of snow samples within the same snow horizon with dirty layer, another snowpack sample (#4') was collected at ca. 1 m away from the location of sample #4 in 2009. Because the thickness of dirty layers in snow pit sequence is ca. 10 cm or more, brown-colored particles are deposited together with snowflakes during snow precipitation rather than dry deposition." (Please see Page 3, Lines 5-12.)

*5) Page 3 line 13 "In order to evaluate the homogeneity of snow samples within the same snow horizon with dust layer, ..." Not sure what is meant by this statement. Need clarification.*

**Response:**

The sentence; "In order to evaluate the homogeneity of snow samples within the same snow horizon with …" has been changed to "In order to evaluate the consistent distribution of snow samples within the same snow horizon with …". (Page 3, Lines 9-10).

*6) Page 3 line 25 "The data of inorganic ...." In addition to back trajectory and lidar data, were mineral, or crustal elemental fractions in the snow contaminants compared with reference material for the different Asian dust regions to verify dust from specific regions?*

**Response:**

We apologize for this error. The sentence; "The data of inorganic ions and trace elements in the reference samples are reported elsewhere (Nishikawa et al., 2000, 2013)." has been revised to "The detailed information of reference samples are reported elsewhere (Nishikawa et al., 2000, 2013)." (Page 3, Lines 21-22).

*7) Page 3 line 30 Why and how was the pH of the samples adjusted to 8.5 to 9.0? If this is described in Kawamura and Kaplan 1984, then including it here just raises questions and isn't informative. This is again stated on page 4 line 4 without explanation.*

**Response:**

Melt snow samples from the Muroro-Daira were slightly acidic. To avoid the evaporative loss of volatile organic acids from the water samples during analytical procedure, the samples were adjusted to pH = 8.5-9.0 with 0.05 M KOH solution.

We have modified the following sentence in response to the above comment.

The sentence; "150 ml of melted snow samples were transferred to a pear-shape glass flask (300 ml) and the pH was adjusted to 8.5–9.0 with 0.05 M KOH solution." has been revised to "150 ml of melted snow samples were transferred to a pear-shape glass flask (300 ml). To avoid the evaporative loss of volatile monocarboxylic acids from samples during analytical procedure, pH was adjusted to 8.5–9.0 with 0.05 M KOH solution to form organic acid salts (e.g., $CH_3COO^-K^+$)." (Page 3, Lines 25-27).

*8) Page 7 line 25 "Although the alkalinity of snow pit samples can be affected…were slightly acidic." Not sure what the relevance of this statement is. I think it is the use of "although" that is throwing me off.*

**Response:**

We are sorry for the unclear description. We have deleted the following sentence from the revised MS.

"Although the alkalinity snow pit samples can be affected by titration of alkaline dust particles, melt snow samples from the Murodo-Daira were slightly acidic."

**Some, but not all grammatical clean-up:**

*9) Abstract Line 17 remove "being" before consistent.*

**Response:**

The term remain the same.

*10) Page 1 line 30, comma after Kawamura citation.*

**Response:**

Based on the suggestion, we have added "," after (Kawamura et al., 2000). (Page 2, Line 1).

*11) Page 2, line 8 have a variety of sources (insert of)*

**Response:**

Based on the suggestion, the phrase; "Formic and acetic acids have variety sources such as primary emission from motor exhausts (Kawamura et al., 2000)…" has been revised to "Formic and acetic acids have a variety of sources such as primary emission from motor exhausts (Kawamura et al., 2000)…". (Page 2, Line8).

*12) Page 5 line 7 insert "the" before laser.*

**Response:**

Based on the suggestion, the sentence "The observation wavelength of laser is 532 nm." has been revised to "The observation wavelength of the laser is 532 nm." (Page 5, Lines 6-7).

*13) Page 7 line 9 change has to was before "involved"*

**Response:**

Based on the suggestion, the sentence; "… Asian Continent has involved with a heavy snow precipitation." has been revised to "… Asian Continent was involved with a heavy snow precipitation." (Page 7, Line 9).

*14) Check uses of "although" and "however", the authors use these two conjunctions are used a lot and not always appropriately.*

**Response:**

Based on the suggestion, we modified following sentences.

The sentence; "However, concentrations of lactic and glycolic acids are 1 and 2 orders of magnitude lower than those of major monocarboxylic acids ($C_1$ and $C_2$), respectively." has been revised to "Concentrations of lactic and glycolic acids are 1 and 2 orders of magnitude lower than those of major monocarboxylic acids ($C_1$ and $C_2$), respectively." (Page 5, Lines 30-31).

The sentence; "Although the pathways of microbial production of branched chain monocarboxylic acids and lactic acid may be different, this strong correlation indicates

that these organic acids are closely linked in the biosynthetic processes associated with bacterial activity in soils." has been revised to "This strong correlation suggests that these organic acids are closely linked in the biosynthetic processes associated with bacterial activity in soils." (Page 9, Lines 20-21).

The sentences "Although bacteria species responsible to branched monocarboxylic and lactic acids have not been reported in the Tateyama snow samples (Maki et al., 2014), our results suggest that branched chain monocarboxylic acids are produced by bacterial process in soils of the Asian Continent and transported over the Japanese Islands with Asian dust. However, contribution of biogenic monocarboxylic acids is much lower than anthropogenic monocarboxylic acids." have been revised to "Bacteria species responsible to branched monocarboxylic and lactic acids have not been reported in the Tateyama snow samples at this time. Our results suggest that branched chain monocarboxylic acids may be produced by bacterial process in soils of the Asian Continent and transported over the Japanese Islands with Asian dust. Contribution of biogenic monocarboxylic acids is much lower than anthropogenic monocarboxylic acids." (Page 9, Lines 25-28).

We have deleted following sentence from the revised MS; "However, it was detected in our snow pit samples."

In addition, we have modified the following sentences.
The sentence "Details of analytical procedure were described previously (Kawamura et al., 2012)." has been revised to "Details of analytical procedure were described previously except for the pH adjustment with KOH solution (Kawamura et al., 2012)." (Page 4, Lines 16-17).

The sentence "They are adsorbed on the pre-existing particles via atmospheric titration with alkaline Kosa particles during the long-range atmospheric transport over the Japanese Islands." has been revised to "They are adsorbed on the pre-existing alkaline Kosa particles via atmospheric titration during the long-range atmospheric transport over the Japanese Islands." (Page 9, Lines 4-5).

The phrase "Total MCA-C/DOC ratio (av. 21%) in 2009 is significantly higher than

those reported in rainwater from Los Angeles …" has been revised to "Total MCA-C/DOC ratio (av. 21%) in 2009 is significantly higher than those reported in rainwater samples from Los Angeles …" (Page 9, Lines 8-9).

The phrase "… suggesting that entrainment of organic acids in snow flakes is significant during the atmospheric transport from China to Japan." has been revised to "… suggesting that entrainment of organic acids in alkaline dusts and snow flakes is significant during the atmospheric transport from China to Japan." (Page 9, Lines 12-13).

The phrase "… secondary photochemical oxidation of anthropogenic toluene, indicating that … " has been revised to "… secondary photochemical oxidation of anthropogenic toluene and other aromatic hydrocarbons, indicating that …" (Page 10, Lines 5-6).

---

## Author Response (AR2)

**Dear Dr. Ryan Sullivan**

**We appreciate the helpful comments made by editor. We have modified the manuscript accordingly. Please see our responses below.**

Below, we indicate in detail the revisions made to the manuscript.

Changes in the revised manuscript are shown in red.

Based on the comment 8, figure numbers were reorganized in the revised MS.

**Comments to the Author:**

The authors have revised the manuscript to largely address the questions and comments raised by the Referees. Importantly, the highly speculative discussion of monocarboxylic acids enhancing the ice nucleation properties of the Asian dust particles has been removed. The manuscript should now be suitable for publication in ACP. There are however, several areas requiring further clarification and revision before the manuscript can be accepted for publication, as listed below. Please address each of these carefully. The manuscript also requires careful copy-editing to bring it up to proper English standards. The list of typos and other corrections below is not a complete list of all the grammatical errors.

*1) Referee 1 asked for information regarding the function used to perform the various regressions, but this detail was not provided in the Response or revised manuscript. I believe they are asking what regression equation was used, e.g. the standard Pearson correlation, or the Demming equation, or another function. The Pearson correlation allows for uncertainty only in the y-variable, while Demming accounts for uncertainty in both the x and y variable. Therefore, the Demming regression seems more appropriate for the regression analysis performed here, as there is uncertainty in both variables. Please also provide the uncertainty for the variables plotted and used in each regression.*

**Response:**

We are sorry for the unclear description. In our previous MS, we used the "standard Pearson correlation (regular linear regression)" for the regression analysis. As suggested, "Deming linear regression" is more appropriate than regular linear regression because of minimization of the distance between the observed data and fitted line in both x and y directions. We used the "Deming linear regression" for the regression analysis in the revised MS. We have modified Figs. 5-10 with the captions as below, except for Figure

7. Please see figures.

Figure 7 shows ion balance. The slope of regular linear regression (y-intercept is zero) shows the balance of ions (cations (x-axis) and anions (y-axis)). Regular linear regression is appropriate than Deming linear regression for this type of data.

Page/line references are used for the following:
*2) 2/5: "as CCN and ice nuclei (IN)." should be deleted, as wet deposition also scavenges aerosol particles, and water soluble gases. The individual contribution of CCN, IN, aerosols, and gases to material found in the snow pack cannot be determined. "Thus, organic acids are scavenged by wet deposition from the upper troposphere." is a more appropriate description.*
**Response:**
As suggested, we deleted the phrase of "as CCN and ice nuclei (IN)". Please see page 2 line 5.

*3) 4/1 & 4/8: should be "blow-down"*
**Response:**
Corrected. Please see page 4 line 1 and page 4 line 8.

*4) 4/23: should be "were injected into an ion chromatograph"*
**Response:**
Corrected. Please see page 4 line 23.

*5) 5/24: "We found that differences in the concentrations of each monocarboxylic acids between sample #4 and sample #4' are comparable to the relative standard deviation." It is not clear what relative standard deviation you refer to here. Of standards analyzed? Please clarify.*
**Response:**
We are sorry for the unclear description. We have added the following description in the revised MS.

"We found that differences in the concentrations of each monocarboxylic acids between sample #4 and #4' are comparable to the total relative standard deviations based on triplicate analysis of real samples." Please see page 5 lines 24-25.

*6) 6/2: "Higher concentrations of monocarboxylic acids were observed in the snow samples with the dust layers than those without dust layers in both 2009 and 2011." Please state the values/range for the samples with versus dust layers to make a quantitative comparison. As the association of dust with the monocarboxylic acids is the major focus of this paper this data should be presented more clearly.*

**Response:**

Based on the comment, we have rephrased to the following sentences in the revised MS.

"Average concentrations of total monocarboxylic acids in the dust layers (2009: 739 ng g$^{-1}$, 2011: 114 ng g$^{-1}$) were greater than that in the without dust layers (2009: 313 ng g$^{-1}$, 2011: 43 ng g$^{-1}$) (Fig. 4)." Please see page 6 lines 3-5.

*7) 6/14: This single-sentence paragraph should be included in one of the neighboring paragraphs and not left on its own.*

**Response:**

Based on the comment, the sentences "The pH of melt snow samples ranged from 4.4 to 6.9 (Table 3). The higher pH were found in samples #1, #3, and #4 (pH = 6.7–6.9), in which dust layers were observed." have merged with the previous paragraph. Please see page 6 lines 14-15.

*8) Results: A large amount of the measurement data are presented as text in the paper, which makes it hard to really grasp the main features of the dataset. Please provide more of the data in the form Figures, as possible, instead of just listing long strings of measured values in the text. A well designed Figure could nicely show the differences in the LWM carboxylic acid concentrations for the different years, and for layers with and without suspected significant dust contributions, for example.*

**Response:**

Based on the comment, we have added Figure 4 as below.

[Figure]

Figure 4. Concentrations of selected low molecular weight monocarboxylic acids in Mt. Tateyama snow samples.

In addition, we have modified in Tables 1, 2, and 3. Shaded columns represent dust layers.

*9) Figure 2: What does the color scale correspond to? Please properly label this on the Figure and explain in the Figure caption. "means no data of dusts" does not make sense, please reword.*

**Response:**

We have modified in Figure 2 caption and added the right y axis label of "Extinction coefficient of dust particles".

[Figure]

Figure 2. Example of lidar measurements of dusts obtained at Imizu, Toyama (ca. 40 km northwest of Mt. Tateyama) during December 1-31, 2008. The color scale indicates extinction coefficient of dust particles based on lidar measurements. Black line represents clouds and gray shade above the black lines represents no data.

*10) Figure 3: "Lines indicate trajectories the snow pit samples with dust layers." does not make sense, please reword.*

**Response:**

Based on the comment, we have rephrased to the following sentence in the revised MS.

"Color lines show the trajectories associated with dust layers as observed by a lidar." Please see Figure 3 caption.

*11) Figure 5: y-axes – is this the natural logarithm of the values? The axes labels say "ln". I think these were changed from log-log to linear plots, and these axes labels need to be corrected.*

**Response:**

Y-axis is the natural logarithm of the values. Based on the comment, we have modified the y-axis label.

[Figure]

Figure 6. Scatter plots of natural logarithm of formic plus acetic acids and pH, and natural logarithm of nss-$Ca^{2+}$ and pH. The solid and dotted lines represent the Deming linear regression.